# Re-Thinking the World with Neutral Monism:Removing the Boundaries Between Mind, Matter, and Spacetime

**DOI:** 10.3390/e22050551

**Published:** 2020-05-14

**Authors:** Michael Silberstein, William Stuckey

**Affiliations:** 1Department of Philosophy, Elizabethtown College, Elizabethtown, PA 17022, USA; 2Department of Philosophy, University of Maryland, College Park, MD 20742, USA; 3Department of Physics, Elizabethtown College, Elizabethtown, PA 17022, USA; stuckeym@etown.edu

**Keywords:** neutral monism, panpsychism, strong emergence, hard problem, explanatory gap, delayed choice quantum eraser, quantum entanglement, conscious observer, no preferred reference frame, constructive explanation, principle explanation

## Abstract

Herein we are not interested in merely using dynamical systems theory, graph theory, information theory, etc., to model the relationship between brain dynamics and networks, and various states and degrees of conscious processes. We are interested in the question of how phenomenal conscious experience and fundamental physics are most deeply related. Any attempt to mathematically and formally model conscious experience and its relationship to physics must begin with some metaphysical assumption in mind about the nature of conscious experience, the nature of matter and the nature of the relationship between them. These days the most prominent metaphysical fixed points are strong emergence or some variant of panpsychism. In this paper we will detail another distinct metaphysical starting point known as neutral monism. In particular, we will focus on a variant of the neutral monism of William James and Bertrand Russell. Rather than starting with physics as fundamental, as both strong emergence and panpsychism do in their own way, our goal is to suggest how one might derive fundamental physics from neutral monism. Thus, starting with two axioms grounded in our characterization of neutral monism, we will sketch out a derivation of and explanation for some key features of relativity and quantum mechanics that suggest a unity between those two theories that is generally unappreciated. Our mode of explanation throughout will be of the principle as opposed to constructive variety in something like Einstein’s sense of those terms. We will argue throughout that a bias towards property dualism and a bias toward reductive dynamical and constructive explanation lead to the hard problem and the explanatory gap in consciousness studies, and lead to serious unresolved problems in fundamental physics, such as the measurement problem and the mystery of entanglement in quantum mechanics and lack of progress in producing an empirically well-grounded theory of quantum gravity. We hope to show that given our take on neutral monism and all that follows from it, the aforementioned problems can be satisfactorily resolved leaving us with a far more intuitive and commonsense model of the relationship between conscious experience and physics.

## 1. Introduction

### 1.1. The Project

In the program opener for the Models of Consciousness Conference devoted to “formal approaches to the mind-matter relation”, held at Oxford in September of 2019, it says the following:
...the subject has generated sustained interest among mathematicians, physicists, and others who aim to translate the results of previous investigations into formal-mathematical models. This interest has been fueled by the observation that many relevant questions, e.g., about the connection between fundamental physics and consciousness, are not amenable to less formal analysis.

This statement and call to arms is historically striking because as many people have noted recently, one way of stating the hard problem of consciousness is that physical phenomena are phenomenally amenable to mathematical modelling and analysis, whereas experiential or qualitative properties are not. Many people have pinned the origin of this erroneous assumption on Galileo who simultaneously committed himself and western culture as it turned out, to the idea that the “book of the universe might be written in entirely mathematical language” and also to the idea that all the properties in the universe can be divided up into primary and secondary qualities ([1], p. 16). Primary properties such as shape, molecular structure, mass, charge, and spin are mind-independent objective properties and thus the purview of mathematical physics, whereas secondary properties such as various sights, sounds, and smells are all subjective properties dependent for their existence on the minds/brains of conscious perceivers, and thus beyond analysis or explanation by mathematical physics. That is, the assumption is that qualia such as “seeing red” are paradigmatically intrinsic and essentially qualitative and thus indescribable formally. This state of affairs led Rebecca Goldstein to say the following ([2], p. 98):
It is somewhat depressing to think of an absolute limit on our science: to know there are things we can never know. ...Mathematical physics has yielded knowledge of so many of the properties of matter. However, the fact that we material objects have experiences should convince us that it cannot, alas, yield knowledge of them all. Unless a new Galileo appears, who offers us a way of getting at properties of matter that need not be mathematically expressible, we will never make any scientific progress on the hard problem of consciousness.

However, contra Goldstein, the idea motivating the Oxford conference and the motivation for this special issue of *Entropy* is that perhaps the Galileo or Galileos of consciousness studies will figure out ways to formally model and explain various aspects of conscious processes and maybe even relate formally some of those aspects to neural processes or more fundamental physics.

This however begs a question. What are the most fruitful and deepest ways to formally model conscious processes and relate them to physics? Of course if one wants to use various formalisms to merely model some feature of conscious experience, there is certainly nothing stopping them. For example, there are those who use the Hamiltonian formalism of quantum mechanics (QM) to model conscious decision making, e.g., thinking thru a decision is like being in a superposition state and making a decision is like wavefunction collapse [3]. There are of course by now many formal models of various aspects of conscious experience, such as using dynamical systems theory and graph theory to model the relationship between brain dynamics and networks and various states and degrees of consciousness. Relatedly, there are various information theoretic models of conscious experience such as Integrated Information Theory (IIT) that purport to explain if not the very existence, at least the content, unity, degree, or types of conscious experience under certain conditions such as waking state, dreaming, psychedelics, anesthesia, etc. [4]. Pragmatically speaking, we do not doubt that there are many such valuable projects going forward, and we say let a thousand flowers bloom.

However, suppose one hopes not to merely formally model conscious experience or seek the neural, dynamical, graphical, information-theoretic, or computational correlates of conscious experience. As Christof Koch puts it ([4], p. 71):
Once science sees the neural correlate of conscious experience face to face, what then? ...But we would still not understand at a conceptual level why this mechanism but not that one constitutes a particular experience. How can the mental be squeezed out of the physical? To paraphrase the New Testament and the philosopher Colin McGinn, how is the water of the brain turned into the wine of experience?

Of course, any formalism such as IIT is open to more than one metaphysical interpretation, thus IIT could be claimed by either strong emergence or panpsychism. Koch clearly prefers the latter, but his general point is clear, we want to work out how mind and matter most fundamentally relate. In the preceding passage Koch is focused just on the brain and biology. Suppose we want to go even further, and find the deepest connection between conscious experience and fundamental physics?

We would argue that the Galileo of consciousness studies will be the one who answers this question. Many now argue that this is where the real Galileo led us astray with his metaphysical distinction between primary/secondary properties which western science then reified into allegedly substantiated empirical fact. As Philip Goff notes in his book *Galileo’s Error* [1], this metaphysical assumption parading as science is a major source of the hard problem to begin with.

If, along with Galileo, one accepts that matter is fundamental and essentially non-conscious and one is a realist about conscious experience, then one is forced into strong emergence and naturalistic property dualism; one is forced to look for something like brute psycho-physical bridge laws or some brute dynamical or causal process from which consciousness emerges [5,6]. The increasingly popular alternative, which agrees that Galileo got it wrong, is some variety of panpsychism which holds that whatever the fundamental entities or processes of physics are, their intrinsic nature is some sort of proto-consciousness or proto-subjectivity, i.e., “what it’s like to be.”

We agree with the panpsychists and the like that Galileo made an error and that one can’t subtract out conscious experience from what we call the “physical” universe. However, there are many ways to reject this assumption, such as neutral monism. With neutral monism, we do not tell a story about how fundamental quantum processes give rise to full-blooded conscious experience (i.e., strong emergence or panpsychism), we tell a story that starts with the nonduality of the so-called mental and physical (neutral monism), why we incorrectly perceive them as distinct, and then we proceed to construct two axioms grounded in that neutral monism that allow us to sketch a derivation of much of physics.

With neutral monism, what we take to be physical and mental are truly nondual, and that means we have to reconceive what the laws of physics, including those of QM, really are. What are such laws? They are, as we will see, constraints on spacetime conceived via neutral monism, i.e., what James calls “the field of experience”. That is, such laws, which are really adynamical global constraints ranging over spacetime, constrain what observers can experience, measure, observe, etc. As we will see, one such constraint is that no observer occupies a preferred frame of reference. This constraint along with one other, turns out to yield a good deal of known physics. Therefore, yes, given neutral monism, “physics”, all of physics, is about “observers” in that they happen to occupy frames of reference (construed broadly), but not in any anthropocentric sense, e.g., conscious observers do not collapse wavefunctions or pop out of wavefunctions collapsing, etc.

What are the most fundamental and universal ontological and formal relationships between conscious experience on the one hand and relativity and QM on the other? Where do we see those relationships? What are the clues? There are many ways to try and tackle these questions. For example, like Kant we could seek the transcendental grounds of the ordinary everyday experience of being a subject in space and time. Only slightly less ambitious, we could seek to use physical experimentation to begin to determine the relationship in question. Think of the Lucien Hardy proposal to see if conscious choice via EEG could violate EPR correlations. Here is what Nicolas Gisin said about Hardy’s idea: “But if someone does the experiment and gets a surprising result, the reward is enormous. It would be the first time we as scientists can put our hands on this mind-body or problem of consciousness” [7].

Engineering and timing problems aside, one could even imagine doing a delayed-choice QM eraser experiment where a person is consciously choosing the state of the slits or the lens directly! Most of us who do QM would bet that a person’s conscious choice could not violate the statistics of QM. If so, this means there are rules of QM dictating individual and intersubjective conscious experience and individual choice, even for people at spacelike separation. What does this suggest about the relationship between conscious experience and QM? It seems unlikely that the mechanism is some sort of non-local “causal” connection. Perhaps it suggests some sort of adynamical global constraint over spacetime, such as a conservation law or the light postulate of special relativity (SR). What would this explanation be like? We will return to this question shortly.

For now, we just want to remind the reader that individual and intersubjective experiences and even conscious decisions are not determined by internal processes alone. Imagine how different our experiences would be if either the light postulate or relativity principle were not true, or if conservation laws regularly failed. Why, we might not have experiences at all in the Kantian sense of the word. Of course, as long as we think of conscious experiences as being qualia generated inside the head, then such laws of physics are at best mere external constraints on experience. Perhaps it is time to start thinking differently. This is where neutral monism parts company with the other accounts we have mentioned so far.

### 1.2. Re-Thinking Fundamental Explanation

In order to fully appreciate neutral monism and our explanatory project some background is in in order. We will start with some more recent background about the state of play in QM. These days many in the quantum information theory end of quantum foundations seek a principle explanation of Bell state entanglement and other features of QM. While reconstructions of QM have been produced, the community does not find them compelling. Many physicists in quantum information theory are calling for “clear physical principles” [8] to account for QM. As Hardy points out [9], “The standard axioms of [quantum theory] are rather ad hoc. Where does this structure come from?” Fuchs points to the postulates of SR as an example of what quantum information theory seeks for QM [8] and SR is a principle theory [10]. That is, the postulates of SR are constraints offered without a corresponding constructive explanation. In what follows, Einstein explains the difference between the two [11]:
We can distinguish various kinds of theories in physics. Most of them are constructive. They attempt to build up a picture of the more complex phenomena out of the materials of a relatively simple formal scheme from which they start out ...
Along with this most important class of theories there exists a second, which I will call “principle-theories”. These employ the analytic, not the synthetic, method. The elements which form their basis and starting point are not hypothetically constructed but empirically discovered ones, general characteristics of natural processes, principles that give rise to mathematically formulated criteria which the separate processes or the theoretical representations of them have to satisfy ...
The advantages of the constructive theory are completeness, adaptability, and clearness, those of the principle theory are logical perfection and security of the foundations. The theory of relativity belongs to the latter class.

It comes as no surprise that in spite of the success of relativity theory as a form of principle explanation, with the light postulate and relativity postulate as fundamental as opposed to some dynamical law, most people assume that, at the end of the day, fundamental explanation in physics and beyond will be constructive, i.e., that it will involve dynamical laws operating over fundamental physical entities, or causal mechanisms of some sort, such as fundamental physical forces. For a book-length attempt to push back against the bias in favor of dynamical and causal explanations in physics see [12]. Therein we argue that not only principle explanations, but other adynamical and acausal global-constraint-type explanations in physics, such as least action principles and Lagrangian-based explanations, are often deeper than their dynamical counterparts.

Theories attempting to explain conscious experience or link it to fundamental physics are no exception in their bias toward dynamical and causal explanation. Indeed, no matter what the theory we are talking about, e.g., Orch OR [13,14,15], emergent panpsychism via “fusion” [16], pseudo-panpsychist-IIT [4], strong emergence/property dualism-IIT, etc., the idea is that matter under some dynamical or causal process, whether physical, computational or information-theoretic, generates unified conscious experiences such as our own [17]. This way of thinking is rarely ever questioned in spite of the hard problems, combination problems, and hard combination problems that inevitably result. What is even more suspect from our point of view, is the idea that all the conscious states of an individual at a time *t* supervene on the brain states of that individual at a time *t*. Whereas, for us, conscious processes are spatiotemporal, and not point-like in space and time. The question that leads to all these problems is simple enough: How do we represent, model, or explain conscious experience formally, starting with fundamental physics as the base? Maybe this way of putting the question is the problem. That is the assumption behind neutral monism anyway.

### 1.3. Overview of the paper

In short, we are going to attempt the following:A principle explanation for the relationship between what we call conscious mind (subjectivity) and what we call physical phenomena (objectivity). This will include an explanation as to why there is an experience of a Dedekind cut between internal mental experience and the external physical world, where none exists absolutely. That is, we will attempt to show that, via principle explanation, there is no longer any mystery about the deepest ontological or formal relationships between conscious experience and physics, and thus no mystery of the experience of subjects in a world of space and time, i.e., no hard problem or explanatory gap.A sketch of a derivation (a principle explanation) of much of known physics from two axioms that constrain and range over the aforementioned neutral events that make up spacetime. These axioms are the formal backbone of our brand of neutral monism and they seamlessly unify subjectivity and objectivity. From these two axioms we will outline a new approach to quantum gravity. We think the quantum information theory community is on the right track in seeking a fundamental principle explanation for QM and indeed all of physics. While they have found many axiomatic bases for QM, they have not produced axioms for physics as a whole, e.g., quantum gravity.We understand that to many readers this project will seem insanely ambitious or just insane. Once you jettison the idea that physics such as quantum mechanics or relativity is fundamental, that property dualism is true and that the deepest explanations are always dynamical or causal-mechanical, all assumptions shared by strong emergence and panpsychism in some form, then there is room to truly re-think the relationship between mind, matter, and spacetime.

In the next section (Section 2) the character and consequences of neutral monism will be more fully fleshed out. That will be the basis for Section 3 wherein our two axioms will be introduced. In Section 4 we show how the axioms explain, in a wholly unified way, the counterintuitive aspects (“mysteries”) of length contraction and time dilation in SR and Bell state entanglement and other mysteries in QM. In Section 5 we show how the axioms, which operate as principle constraints over spacetime, make impossible any disagreement between conscious experience and the QM rules governing any QM set-up in spacetime, such as the Hardy experiment mentioned above, delayed choice quantum eraser and Wigner’s friend. That is, the two axioms constrain not only the behavior of what we label “physical entities”, but also constrain the individual and intersubjective experiences and conscious choices of conscious agents. We conclude with a summary in Section 6.

## 2. Neutral Monism

Just as the work of David Hume, Immanuel Kant, and Ernst Mach was essential for Einstein’s epiphanies behind SR ([18], pp. 82–83), so for us neutral monism (i.e., radical empiricism) not only shows us how to properly situate consciousness in the world formally and metaphysically, but it shows how to move forward in physics as well. Indeed, in many ways these turn out to be the same project.

Here is what Einstein said he gleaned from Hume and Mach, one must eliminate concepts that “have no link with experience such as absolute simultaneity and absolute speed” ([18], p. 131). As James noted every scientific theory and all scientific explanations are, however opaquely, rooted in some metaphysical picture of the world such that “the juices of metaphysical assumptions leak in at every joint” ([19], p. 112). The radical empiricism (neutral monism) of James is just an extension of the empiricism of Hume and others. In the words of Eugene Taylor ([19], p. 130):
Radical empiricism was, nevertheless, psychological; that is to say, it placed immediate experience at the center of everything we have to say about the universe. Consciousness, therefore, knower-and-known, subject-and-object, person-and-world, formed the basis of all science and all knowledge-getting. Positivistic science had to conform as much to the dictates of such psychology, as psychology was trying to conform to such a science.

Radical empiricism is thus a metaphysics and epistemology of science. Taking it on board allows us to reconceive scientific explanation just as Einstein did with his principle explanation in relativity.

The basic idea of neutral monism is that the mental and physical are nondual (not essentially different) and they are neutral in virtue of being neither essentially mental nor physical, and in virtue of the fact that they are grounded in something neutral. This ubiquitous “something” does not “pervade” spacetime, it does not “generate” spacetime, it is co-extensional with spacetime. According to neutral monism, our conscious experience of an external world is not some virtual model or construction of the world trapped in the mind and generated by the brain, while the actual external physical world lies outside us forever as I-know-not-what noumena. In neutral monism there is only the spatiotemporally extended world of experience, of which we are a part. It will become clear as we go along what all this entails about the relationship between physics and conscious experience.

Here is how James describes neutral monism:
A given undivided portion of experience, taken in one context of associates, play[s] the part of the knower, or a state of mind, or ‘consciousness’; while in a different context the same undivided bit of experience plays the part of a thing known, of an objective ‘content.’ In a word, in one group it figures as a thought, in another group as a thing([20], p. 533).
The relation itself is a part of pure experience; one if its ’terms’ becomes the subject or bearer of the knowledge, the knower, the other becomes the object known([20], p. 534).
Things and thoughts are not at all fundamentally heterogeneous; they are made of one and the same stuff, stuff which cannot be defined as such but only experienced; and which one can call, if one, wishes, the stuff of experience in general([21], p. 110).
“Subjects” knowing, “things” known, are “roles” played. Not “ontological” facts([21], p. 110).

From the perspective of neutral monism, the claim that the world is carved at the joints a la physical/mental; inner/outer; subject/object, etc., is not a datum, but rather an inductive projection. As James puts it, “Subjectivity and objectivity are affairs not of what an experience is aboriginally made of, but of its classification” ([22], p. 1208). Allegedly “inner” experience is not inherently or essentially mental and the so-called “external” world is not inherently non-mental or physical. “Pure experience” (as James calls it), in itself, “is no more inner than outer. ...It becomes inner by belonging to an inner, it becomes outer by belonging to an outer, world” [23].

Neutral monism provides a truly neutral and fundamental base for which so-called physical properties and qualitative properties are but nondual aspects. Neutral monism is fully monistic. No dualism of any sort remains. “It is only in retrospect, when pure experience is ‘taken,’ i.e., talked of, twice [theorized about inductively or reconceived], considered along with its two differing contexts respectively, by a new retrospective experience after the fact that the same indivisible portion of experience assumes the character of subject and object, knower and known” ([24], p. xxiv). Any hint of dualism is therefore a cognitive illusion or inductive projection.

As James scholars have often noted, his “views were not well received or accurately interpreted” in his own time ([24], p. xi). Some have even portrayed James’ view as a kind of eliminativism or behaviorism because he says things of this nature, “Consciousness, as it is ordinarily understood, does not exist” ([21], p. 109). What we hope is clear to the reader now is that James is not denying the existence of conscious experience as such, but only a particular conception of consciousness. Namely, he is rejecting the idea of consciousness as qualia (inner tropes of experience that could exist without a subject as something over and above subjectivity). People often fail to appreciate this point because they leave out the second half of the preceding quote, “any more than does matter” ([21], p. 109). Taking the quote in full, we see that James is really rejecting the primary/secondary property distinction and the idea that matter is a substance with essentially, intrinsic physical properties. Unlike panpsychism, James is not replacing intrinsic physical properties with essentially qualitative ones such as qualia or subjectivity.

In neutral monism, what we call spacetime is nothing but the events therein and those events are neither inherently mental nor inherently physical. Russell calls such “neutral” 4D finite events “unstructured occurrences,” such as “hearing a tyre burst, or smelling a rotten egg, or feeling the coldness of a frog” ([21], p. 287). Contrast this with panpsychism where what is fundamental is both inherently physical and mental. Russell goes on to say that, “Matter and motion ...are logical constructions using events as their material, and events are therefore something quite different from matter in motion” ([21], p. 292). As he puts it, “It is a mere linguistic convenience to regard a group of events as states of a ’thing’, or ’substance’, or ’piece of matter’, or a ’precept”’ ([22], p. 284). Going further he says, “electrons and protons ...are not the stuff of the physical world” ([21], p. 386). Again, “bits of matter are not among the bricks out of which the world is built. The bricks are events, bits of matter are portions of the structure to which we find it convenient to give separate attention” ([23], p. 329).

As Russell puts it, “The whole duality of mind and matter ...is a mistake; there is only one kind of stuff out of which the world is made, and this stuff is called mental in one arrangement, physical in the other” ([24], p. 15). Compare this to the words of William James, “Things and thoughts are not fundamentally heterogeneous; they are made of one and the same stuff, stuff which cannot be defined as such but only experienced; and which one can call, if one wishes, the stuff of experience in general. ...‘Subjects’ knowing ‘things’ known are ‘roles’ played, not ‘ontological’ facts” ([25], p. 63).

The point is that when Galileo and others insisted on the primary/secondary distinction or any other kind of dualism of inner mental experience and external physical world, they doomed us to the mind/body problem and the hard problem. Such dualisms are not empirical data we experience directly, they are cognitive illusions, inductive projections of theorizing minds. The way to undo this mistake is not to merely move the location of the mysterious dualism, i.e., qualia and subjects springing from from brains (strong emergence) to fundamental physics, i.e., qualia and subjects hiding in basic matter (panpsychism), but to reject completely and thoroughly the primary/secondary property distinction with neutral monism.

James says, neutral monism is “a radically pluralistic monism of pure experience” ([26], p. xiv). James’ expression “pure experience” is not meant to invoke sense data theory or idealism, it’s his expression for what Russell calls “events.” All that exists is pure experience and pure experience is all that exists according to James. This is what makes James’ view monistic. It is also “radically pluralistic” in that “pure experience” is infinitely varied in its nature. It is as James says, simply “made of that, of just what appears, of space, of intensity, of flatness, of browness, of heaviness, or what not.” It is “the instant field of the present ...plain, unqualified actuality or existence, a simple that.” ([26], p. xv).

Events, such as everyday experiences of flat tires, concerts, conversations, etc., which are neither essentially physical nor mental, are fundamental and exhaustive, and they are grounded in and one with the neutral “Presence”—what James calls the “instant field of the present." Neutral monism is a rejection of any sort of atomism and the primary/secondary distinction. Reality is fundamentally a neutral network of extrinsic dispositions or relations. The existence of all things and their properties is metaphysically and physically interdependent in a multiscale fashion, and this interdependence is also spatiotemporal, i.e., 4D. In metaphysical parlance, this is a rejection of the hierarchy or foundationalism thesis. Relations between smaller and larger scales are not anti-symmetric, transitive, or anti-reflexive. This fundamental contextuality built into neutral monism will later be part of what leads us to a non-hierarchical way of relating the classical and the quantum.

The neutral base in question is what James calls “unqualified actuality” and “the instant field of the present”. We like to call it Neutral Pure Presence (i.e., “pure being”) or “Nowness”. Philosophers and physicists have long noted that there is something special about the present moment or the Now. Carnap quite famously has quoted Einstein as having said that, “The problem of the Now worried him seriously. The experience of the Now means something special for man, something essentially different from the past and the future, but that this important difference does not and cannot occur within physics. There is something essential about the Now which is just outside of the realm of science” [25]. What Einstein was alluding to in his discussion with Carnap is exactly what James posits as fundamental, it is pure being itself, and it is only outside the realm of science in the sense that it is fundamental.

To paraphrase Wheeler and Hawking, Neutral Pure Presence is “what puts the fire in the equations”. Here is how Wheeler puts the matter, “Explain time? Not without explaining existence! Explain existence? Not without explaining time! To uncover the deep and hidden connection between time and existence ...is a task for the future” ([26], p. 119–120). Wheeler got it exactly right, time (as in Nowness) and existence are indeed one and the same. Neutral Pure Presence is one with, yet not exhausted by, its various nondual aspects, i.e., what James calls “the instant field of the present” and what Russell calls “events”.

If all of this strikes you as a crazy idea, for what it is worth, plenty of other non-mystical types are taking its possibility seriously these days. Here is how Goff expresses the idea: “Formless consciousness is the intrinsic nature of spacetime itself, in a way that is not localized but equally present at all regions of spacetime” ([1], p. 208). As Goff notes, the implication here is that “spacetime is not some great container that physical objects are located in. At a fundamental level, all that really exists is spacetime” ([1], p. 208). Christof Koch talks about the “pure experience”, “pure presence”, or “naked awareness” state associated with certain meditative practices ([4], p. xiii). Koch goes on to note that such “mystical experiences” are common to many religious traditions and are “characterized as having no content: no sounds, no images, no bodily feelings, no memories, no fear, no desire, no ego, no distinction between the experiencer and the experience, the apprehender and the apprehended (nondual)” ([4], p. 7). As we will discuss at length in the next section, one important thing to note about such experiences (as well as others such as emerging from the deep sleep state), is that subject/object, self/world in-space-and-time, always disappear together and co-arise together.

However, Goff is quite wrong in his suggestion that this view is a form of panpsychism or panqualityism, at least as Russell, James, and ourselves conceive of neutral monism. Panpsychism is committed to the co-fundamentality of material and mental properties, and by definition associates fundamental mentality with whatever is physically fundamental; and the panquiddities or panqualityism view is committed to the idea that, “the qualities that we find in experience can exist unexperienced and they do exist in fundamental matter” ([27], p. 160), not so in the case of neutral monism. Panpsychism but not neutral monism, remains committed to the “physicalist position ...that any metaphysically basic facts or laws—any unexplained explainers, so to speak—are facts or laws within [fundamental] physics itself” ([28], p. 560).

Neutral monism, unlike panpsychism, is a true monism wherein neither material nor mental properties are fundamental. What Goff calls “formless consciousness”, what we call Neutral Pure Presence is not a feature of fundamental physical entities, it is more fundamental than and yet one with what we call “physical entities”. Rather than say “formless consciousness is the intrinsic nature of a physical entity: spacetime” ([1], p. 208), one should say that spacetime is grounded in and one with Neutral Pure Presence and the latter is the only thing that is metaphysically autonomous; all other properties are completely extrinsic, and there are no such things as essentially physical or mental properties. Nothing remains of the problematic primary/secondary property distinction, thus fully correcting “Galileo’s error” in a way fully consistent with science. Both so-called physical properties that we relate to an “external” world and so-called qualitative properties that we relate to an “inner” world of the mind, being nondual, are fully relational. With this new neutral monist understanding of the world, let us proceed to a new conception of physics grounded therein.

## 3. Neutral Monism and the Axioms of Physics

### 3.1. Background

In order to appreciate our axioms and how they undergird the physics going forward, some background into the history of ideas will be helpful. Here is the important background for the first of our two axioms of physics. For Kant, given his unity of apperception, time is an a priori condition for experience, no subjectivity, no time or space. Kant here is providing a transcendental analysis in mentalistic terms. This means that the dynamical character of thought/experience and the world are two sides of the same coin. Kant’s transcendental arguments from *The Critique of Pure Reason* are supposed to show that we must conceive of the world in a certain way, structure it internally according to certain categories such as time, space, and causation. Those arguments are fraught with many interpretative perils and controversies, but the basic idea is that experience is possible for me only if some experiences are conceptualized as being of enduring objects, enduring through time and space. Likewise, to experience a world of enduring objects there must be some sense of an enduring self. You cannot have one without the other ([29], Chap. 24).

Kant says time cannot be perceived in itself yet one cannot perceive events without spatial arrangements and temporal coordination. Time and space are necessary a priori conditions of the possibility of experience. As he puts it, space and time are pure forms of intuition. Space is the form of outer sense, time the form of the inner sense. This is to say that in defining the notion of an object or thing one must include the concept of time and space, and vice-versa. The concepts of space, time, and thing can’t be defined in isolation without bringing in the other concepts, they are only co-definable. Likewise, the idea of an enduring subject or perceiver can only be co-defined with respect to an enduring world or object that is being perceived and vice-versa.

Here is James’ ultimate statement about Kant’s transcendental idealism and its relation to radical empiricism ([20], p. 535):
If neo-Kantianism has expelled earlier forms of dualism, we shall have expelled all forms if we are able to expel neo-Kantian in its turn. For the thinkers I call neo-Kantian, the word consciousness today does no more than signalize the fact that experience is indefeasibly dualistic in structure. It means that not subject, not object, but object-plus-subject is the minimum that can actually be. The subject-object distinction meanwhile is entirely different from that between mind and matter, from that between body and soul. Souls were detachable, had separate destinies; things could happen to them. To consciousness as such nothing can happen, for, time itself, it is only a witness of happenings in time, in which it plays no part.

James makes it explicit that he thinks that Kant is on the right track but he wants to be rid of the inherent dualisms in Kant such as that between phenomena/noumena, consciousness and content, etc., that is what radical empiricism (neutral monism) is all about. As James puts it, “that the only things that shall be debatable among philosophers shall be things definable in terms drawn from experience” ([30], p. xii). James is rejecting any reified notion of consciousness or self as existing apart from the content of experience. James says that the reification of self only arises when “a given ’bit’ is abstracted from the flow of experience and retrospectively considered in the context of different relations, relations that are external to the experience taken singly but internal to the general flow of experience taken as a whole”. For James, “the dualism of knower and known is an external dualism of experienced relations not an inner dualism of substance” ([24], p. xvi).

Kant and the others are right that we do not experience things in time and space but rather we experience them temporally and spatially. Given neutral monism they are wrong to say that this is an imposition of individual minds and their categories upon some unknowable noumenal world (the thing-in-itself). The mistake, in one form or another, which we find in both the analytic and continental tradition, is representationalism. Once we take neutral monism on board we can immediately see that there is no need for (and no sense for) representationalism to explain the experience of enduring subjects and objects in space or time. The point here is that subject and object co-exist as a subject in a world in space and time, so the agent is not trapped behind “a veil of perception” but is directly part of the world, and the external world is not some noumenal container onto which the subject projects a virtual reality. Given neutral monism, (transcendental) phenomenology cannot be and should not be divorced from natural science, and experience cannot be separated/bracketed from the natural world, as there is no primary/secondary distinction. As James puts it, “Consciousness connotes a kind of external relation, and does not denote a special stuff or way of being” [20]. Neutral monism entails a kind of direct realism in that the thing represented and the thing being represented are one in the same. Perception is direct and immediate “acquaintance” with the world.

Following Kant’s unity of apperception, which you will notice is a principle explanation itself, we can go a step further: it is only when the subject/object division exists that one gets a world in space and time. Subject/object and world in space/time are both necessary and sufficient for one another. Phenomenology in the west and many Asian traditions such as Hinduism and Buddhism have noted that self/world, subject/object, always appear and disappear together as in going in and out of deep sleep, deep meditative states, and certain psychedelically induced states.

As James says, “As ‘subjective’ we say that the experience represents; as ’objective’ it is represented” ([31], p. 480). Indeed, there is only “the instant field of the present. ...It is only virtually or potentially either object or subject as yet. For the time being, it is plain, unqualified actuality or existence, a simple that” ([31], p. 482). It is only the discursive intellect that later creates or projects these dualisms in an inductive act of interpretation. Keep in mind this discursive intellect is not some a priori cognitive category through which noumena is filtered. The mind’s inference to the dualism of knower/known, subject/object, etc., happens after the fact.

James’ answer to an objection as to the impossibility of thoughts and things being nondual will help us see that he agrees with Kant about the relationship between time, space, objecthood, and subject ([31], p. 489):
If it be the self-same piece of pure experience, taken twice over, that serves now as thought and now as thing–so the objection runs–how comes it that its attributes should differ so fundamentally in the two takings. As thing, the experience is extended; as though, it occupies no space or place. As thing it is red, hard, heavy; but who ever heard of a red, hard or heavy thought? Yet even now you said that an experience is made of just what appears, and what appears is just such adjectives. How can the one experience in its thing-function be made of them, consist of them, carry them as its own attributes, while in its thought-function it disowns them and attributes them elsewhere. ...To begin with, are thought and thing as heterogenous as is commonly said? Their relations to time are identical. ...Sensations and apperceptive ideas fuse here so intimately that you can no more tell where one begins and the other ends.

James’ neo-Kantian point is that what we call the “external” flow of time, so-called “things” changing, and the “internal” flow of time, so-called “thoughts” and “feelings” changing, is one and the same experience. We do not need any Kantian categories to explain this experience of change. As James puts it, “The ‘I think’ which Kant said must be able to accompany all my objects, is the ’I breathe’ which actually does accompany them. ...That entity [consciousness] is a fiction, while thoughts in the concrete are fully real. Thoughts in the concrete are made of the same stuff as things are” ([20], p. 543). As he says, “the parts of experience hold together from next to next by relations that are themselves parts of experience” ([20], p. 534), no internal cognitive filters required. As he puts it, “According to radical empiricism, experience as a whole wears the form of a process in time” ([20], p. 540). Of course, James is not denying that there are real and important differences between thoughts and things. You can lift the latter but not the former.

What then is the self or subject for James? It is just this [32]:
The individualized self, which I believe to be the only thing properly called self, is a part of the content of the world experienced. The world experienced (otherwise called the “field of consciousness”) comes at all times with our body as its centre, centre of vision, centre of action, centre of interest. Where the body is is “here”, where the body acts is “now”; what the body touches is “this”; all other things are “there”, and “then” and “that”.

To provide the necessary historical background for our second axiom, let us paraphrase from Einstein’s bedrock conception of the enterprise of physics [33], quoting phrases and terms from his text. Physics is the study of “bodily objects” moving in 3-dimensional space as a function of time under the influence of their mutual forces (“the statement of a set of rules”). As Einstein pointed out, there are already some assumptions there, so it is best to start with “all sense experiences”. I am the spatiotemporal origin of “all sense experiences”. I assume a subset of “all sense experiences” represents other perceivers. For example, my perception of you is a subset of all “sense experiences”, so I will assume you also have “sense experiences”. In Einstein’s words, “partly in conjunction with sense impressions which are interpreted as signs for sense experiences for others”. Therefore I am the spatiotemporal origin of “*my* sense experiences”, i.e., I am just one perceptual origin (PO). I communicate with other (human) perceivers to construct a model of objective/physical reality (the “real external world”) that reconciles the disparate, but self-consistent (see below) elements of our “sense experiences”. In Einstein’s words, “the totality of our sense experiences ...can be put in order”.

We then use this model to explore regularities and patterns in the events we perceive. We mathematically describe these regularities and patterns and explore the consequences (experiments). In Einstein’s words, “operations with concepts, and the creation and use of definite functional relations between them, and the coordination of sense experiences to these concepts”. We then refine our model of physical reality as necessary to conform to our results. This allows us to explain the past, manipulate physical reality in the present (to create new technology, for example), and to predict the future. While defining physics all the way down to individual “sense experiences” may seem unnecessarily detailed, it is crucial to understanding the relationship between consciousness and physics being proposed here. In turn, we think that understanding will help the reader see how it could be the case that often the best explanations in physics are principle explanations, or explanations in terms of adynamical global constraints on sense experiences.

Putting this all together we can say the following. From our take on neutral monism we understand that each subject is just a point of origin (PO) of Neutral Pure Presence. The perceptions of each PO form a context of interacting trans-temporal (enduring) objects (TTOs) for that PO. Since TTOs are “bodily objects” with worldlines in spacetime, TTOs are coextensive with space and time, i.e., there is no “subject” without “object”. When POs exchange information about their perceptions, they realize that some of their disparate perceptions fit self-consistently into a single spacetime model with different reference frames for each PO. Thus, physicists’ spacetime model of the “Physical” represents the self-consistent collection of shared perceptual information between POs, e.g., perceptions upon which Galilean or Lorentz transformations can be performed.

Of course, not all perceptions are amenable to modeling in the spacetime of physics. For example, one member of a group of observers dining together might see a dancing pink elephant on the dinner table that no one else sees. In this case, the dancing pink elephant is not a TTO in our model of the Physical (the “real external world” per Einstein) because it is not self-consistent with the perceptions of the other POs in that spacetime region. That is not to say the perception is not “real”, since all perceptions are subsets of Neutral Pure Presence by definition and are therefore equally “real” in that sense. The study of these perceptions falls outside the purvey of physics by definition, i.e., to model and explore regularities and patterns in the self-consistent collection of shared perceptual information between POs.

This way of thinking about physics is obviously not unique to us, but it is not even unique to the likes of Einstein, Mach, James, and other radical empiricists. Thinkers such as Hermann Weyl and Arthur Eddington who were responsible for deepening and advancing general relativity (GR) among other things, were very much influenced by neo-Kantianism (Kantian modes of explanation minus the categories and the noumena) and Husserlian phenomenology, which is also a a kind of neo-Kantian project. This way of thinking led Weyl and others to attempt to continue what they thought was Einstein’s revolutionary project of the “geometrization of physics”. Here is how Thomas Ryckman describes that project ([34], p. 195):
The desired conception of a completely impersonal world is only expressible as a geometrical structure. Thus relativity theory (taken as including something like Weyl’s extension of the class of conceivable observers) has completely overturned the older conception of an external world as substance or material.

Here is how Weyl himself puts it: Physics is the “Construction of objective reality out of the material of immediate experience” ([34], p. 117).

Weyl is basically suggesting that spacetime is just the relationships or possible relationships between POs and their perceptions. What is called “objective reality” is what is common to all POs at least in potentia, and mathematical physics is just the codification of those relationships. It should be understood that the focus here is on various invariances across the perceptions of POs and the various adynamical global constraints on those perceptions that enforce those invariances. As Eddington puts it, “physics is about the world from the point of view of no one in particular” ([34], p. 195). Thus, physics is about all the possible perspectives of all POs and their perceptions. For example, consider the role of tensors in GR and their relationship to coordinate systems.

Coordinate transformation is important because relativity states that there is not one reference point (or perspective) in the universe that is more favored than another. As Weyl put it, “The explanation of the law of gravitation thus lies in the fact that we are dealing with a world surveyed from within” ([34], p. 117). Keep in mind that the beauty of neutral monism is that talk about POs and their perceptions should be understood not as some sort of positivism, or some brand of idealism (subjective or otherwise), or merely as bracketed phenomenology, but in terms of James’ “instant field of the present” and what Russell calls “events”. That and that alone is what spacetime is.

### 3.2. The Physics

With that understanding of consciousness and physics, we can posit the following axiomatic basis for physics:Axiom 1: Interacting “bodily objects” coextensive with space and time form the context of all self-consistent, shared perceptual information between POs and these perceptions constitute different reference frames in that spacetime model of the Physical (the “real external world”).Axiom 2: For all of physics, the perceptions of any particular PO do not provide a privileged perspective of the Physical. This is known as “no preferred reference frame” (NPRF).

In the remainder of Section 3, we will survey how current physics follows from these axioms and how they lead to an empirical approach to quantum gravity (Figure 1). In short, physics seeks to model POs’ self-consistent collection of shared perceptual information about interacting TTOs in spacetime. Axioms 1 and 2 constrain POs’ experience of the TTOs of classical physics in spacetime that interact via quantum physics per quantum-classical contextuality (Section 3.2.1). Thus, we first show how classical physics and the gauge fixing per gauge invariance of quantum physics are consistent with quantum-classical contextuality per Axiom 1 (Section 3.2.2). Then we point out how Axiom 2 is related to the notion of symmetries, which represent the fact that there is just one “real external world” harboring many, but always equal perspectives as far as physics is concerned (Section 3.2.3). In order to clarify this model of the Physical, we share our Relational Blockworld interpretation of QM (Section 3.2.4) before bringing it all together into the hierarchy of physics (Section 3.2.5) and our approach to quantum gravity (Section 3.2.6).

#### 3.2.1. Quantum-Classical Contextuality

Notice that Axioms 1 and 2 are constraints upon and range over spacetime, contributing to the experience of a world of TTOs. These axioms are functionally equivalent to Kant’s categories, but they are not constraints imposed by individual minds. Let us now make our argument that the TTOs of classical physics interact via quantum physics establishing quantum-classical contextuality.

Newton’s laws can be written in constraint form as the global conservation of momentum
(1)$$\sum_{i=1}^{N}{\vec{p}}_{i}=\mathrm{constant}$$

*N* is a very large number of course, but in practice we only deal with small subsets under the assumption that these subsets are isolated from their complement to within experimental limits. Notice that we are tacitly assuming an ontology of TTOs, i.e., objects with worldlines in spacetime, as is common throughout physics. As we argued above, TTOs, space, and time are coextensive and fundamental in both our epistemology and ontology per neutral monism (Axiom 1). Continuing, Equation (Equation 1) means
(2)$$\sum_{i=1}^{N}\frac{d{\vec{p}}_{i}}{dt}=0$$

Let us focus on the $N^{th}$ object, rewriting Equation (Equation 2) as
(3)$$\frac{d{\vec{p}}_{N}}{dt}=-\sum_{i=1}^{N-1}\frac{d{\vec{p}}_{i}}{dt}\equiv {\vec{F}}_{\mathrm{ext}}$$

Equation (Equation 3) is Newton’s second law of course and the game of classical mechanics is largely played by articulating some ${\vec{F}}_{\mathrm{ext}}$ then solving the differential equation(s) produced by Equation (Equation 3). However, the articulation of some ${\vec{F}}_{\mathrm{ext}}$ is ad hoc, i.e., the functional forms for the forces of classical mechanics are simply “put in by hand”. Indeed, classical mechanics would allow Equation (Equation 2) to be satisfied trivially by having each $\frac{d{\vec{p}}_{i}}{dt}=0$, so classical mechanics does not say why an individual object experiences any change in its momentum. Nor does it provide any mechanism beyond ad hoc forces to explain the reciprocal changes in momentum between objects in order to satisfy Equation (Equation 3). Again, it could just as well have been the case that the worldlines of objects are always geodesics in Newtonian spacetime such that two different objects would simply pass through one another where their worldlines intersected.

Of course, without exchanges of momentum there can be no exchange of photons, so objects would not be able to “see” each other either. In other words, no reciprocal changes in momentum means no interaction and no interaction means the objects can neither acquire nor exchange information about or with each other. Since that certainly does not represent objective reality, we need another formalism to account for the fundamental nature of interaction/momentum exchange and that formalism is QM. We then understand QM to produce a 4D probability distribution for fundamental momentum exchanges in the classical spacetime context of beam splitters, mirrors, sources, detectors, etc., the average of which is consistent with classical mechanics in accord with conventional wisdom [12].

While there is no fundamental quantum of momentum or energy for the interaction of classical objects (there does not seem to be any limit to the size of a quantum exchange of momentum), there is a fundamental quantum of action *h* for all interactions. Consider, for example, the emission of a photon when a particle in a box falls from an excited state to the ground state. The energy of the photon equals the difference in energy between the excited state and ground state ΔE and is given by hf. Suppose the transition occurs in time Δt, the period of the photon, then we have ΔEΔt=h (not to be confused with the uncertainty principle). This holds for a transition between any two levels and indeed, it holds for any quantum transition in any classical object whether it is a particle in a box or an atomic or molecular transition, etc. Therefore, while the amount of energy involved in the transition and the duration of the transition can take on any values, the amount of action represented by the emission or reception of a photon is always *h*.

We recognize immediately the analogy with SR’s variable spatial displacements and temporal durations combining into the invariant spacetime interval, so that everyone measures the same speed of light *c*, another fundamental constant of Nature (Section 4.1). Indeed, as we will show in Section 4, the invariance in the measurement of *c* per Axiom 2 accounts for the mysteries of time dilation and length contraction, while the invariance in the measurement of *h* per Axiom 2, combined with conservation per Axiom 1, accounts for the QM mystery of Bell state entanglement whence the Tsirelson bound [35,36]. Now let us extend this to the quantum exchange of momentum involving a massive particle.

Suppose a classical object emits/receives a massive particle in a quantum interaction causing it to lose/gain momentum Δp. Further, suppose that this momentum emission/reception occurs over a distance Δx given by the de Broglie wavelength $\lambda_{dB}$ of that particle. Then the action involved in this quantum emission/reception process is again ΔpΔx=h (again, not to be confused with the uncertainty principle). Of course, the photon (massless) case can be put in precisely this form via $\Delta t=\frac{\lambda }{c}$, λ=Δx, and $\frac{\Delta E}{c}=\Delta p$. Thus, we see that every fundamental interaction between classical objects involves the same fundamental unit of action *h*, regardless of the energy, momentum, temporal duration, or spatial extent of the emission/reception events.

#### 3.2.2. Axiom 1 and Quantum Gauge Invariance

Axiom 1 is satisfied with all existing theories of physics in that they all obey the “boundary of a boundary principle”, ∂∂=0, which subsumes gauge invariance and local conservation principles [37,38] (Figure 2).

For example, Einstein’s equations of GR ($G_{\alpha \beta }=\frac{8\pi G}{c^{4}}T_{\alpha \beta }$) are divergence-free (${▽}^{\alpha }G_{\alpha \beta }=0$ and ${▽}^{\alpha }T_{\alpha \beta }=0$), which means you have local conservation of energy-momentum—what flows into a region of space accumulates there or flows out. This is germane to the identification of a TTO through time.

The TTOs of classical physics interact per quantum physics, so the quantum exchange of energy-momentum between TTOs per QM must be consistent with their divergence-free nature. This fact is represented by gauge fixing per gauge invariance. We can illustrate that by using the non-relativistic limit of the Klein–Gordon equation giving the free-particle Schrödinger equation by factoring out the rest mass contribution to the energy *E*, assuming the Newtonian form for kinetic energy, and discarding the second-order time derivative ([39], p. 172).

To illustrate the first two steps, plug $\phi =Ae^{i(px-Et)/\hbar }$ into the Klein-Gordon equation and obtain $\left(-E^{2}+p^{2}c^{2}+m^{2}c^{4}\right)=0$, which tells us *E* is the total relativistic energy. Now plug $\psi =Ae^{i(px-Et)/\hbar }$ into the free-particle Schrödinger equation and obtain $\frac{p^{2}}{2m}=E$, which tells us *E* is only the Newtonian kinetic energy. Thus, we must factor out the rest energy of the particle, i.e., $\psi =e^{imc^{2}t/\hbar }\phi$, assume the low-velocity limit of the relativistic kinetic energy, and discard the relevant term from our Lagrangian density (leading to the second-order time derivative) in going from ϕ of the Klein–Gordon equation to ψ of the free-particle Schrödinger equation. We will simply outline the details here.

Overall, we start with a transition amplitude from quantum field theory to get our generating function whence the propagator that gives us the QM probability amplitude ψ. The transition amplitude for the Klein–Gordon equation is
(4)$$Z\left(J\right)=\int D\phi \;\exp\left[\frac{i}{\hbar }\int d^{4}x\left(\frac{1}{2}{\left(\partial \phi \right)}^{2}-\frac{1}{2}{\bar{m}}^{2}\phi^{2}+J\phi \right)\right]$$
which in (1+1)D is
(5)$$Z\left(J\right)=\int D\phi \;\exp\left[\frac{i}{\hbar }\int dxdt\left(\frac{1}{2}{\left(\frac{\partial \phi }{\partial t}\right)}^{2}-\frac{c^{2}}{2}{\left(\frac{\partial \phi }{\partial x}\right)}^{2}-\frac{1}{2}{\bar{m}}^{2}\phi^{2}+J\phi \right)\right]$$
where $\bar{m}=\frac{mc^{2}}{\hbar }$. Making the changes described above with $\psi =e^{i\bar{m}t}\sqrt{\bar{m}}\phi$ gives the non-relativistic Klein–Gordon transition amplitude corresponding to the free-particle Schrödinger equation ([39], p. 173)
(6)$$Z\left(J\right)=\int D\psi \;\exp\left[\frac{i}{\hbar }\int dxdt\left(i\psi^{*}\left(\frac{\partial \psi }{\partial t}\right)-\frac{c^{2}}{2\bar{m}}{\left(\frac{\partial \psi }{\partial x}\right)}^{2}+J\psi \right)\right]$$

Now integrate the second term by parts and obtain
(7)$$Z\left(J\right)=\int D\psi \;\exp\left[\frac{i}{\hbar }\int dxdt\left(i\psi^{*}\left(\frac{\partial \psi }{\partial t}\right)+\frac{\hbar }{2m}\psi^{*}\frac{\partial^{2}\psi }{\partial x^{2}}+J\psi \right)\right]$$

This gives
(8)$$Z\left(J\right)=\int D\psi \;\exp\left[\frac{i}{\hbar }\int dxdt\left(\frac{1}{2}\psi^{*}K\psi +J\psi \right)\right]$$
where
(9)$$K=\left(2i\frac{\partial }{\partial t}+\frac{\hbar }{m}\frac{\partial^{2}}{\partial x^{2}}\right)$$

Discretizing this on a four-node graph, for example, we find that **K** is given by
(10)$$K=\left(\begin{array}{cccc} \left(\frac{2i}{\Delta t}-\frac{\hbar }{m\Delta x^{2}}\right) & -\frac{2i}{\Delta t} & \frac{\hbar }{m\Delta x^{2}} & 0\\ -\frac{2i}{\Delta t} & \left(\frac{2i}{\Delta t}-\frac{\hbar }{m\Delta x^{2}}\right) & 0 & \frac{\hbar }{m\Delta x^{2}}\\ \frac{\hbar }{m\Delta x^{2}} & 0 & \left(\frac{2i}{\Delta t}-\frac{\hbar }{m\Delta x^{2}}\right) & -\frac{2i}{\Delta t}\\ 0 & \frac{\hbar }{m\Delta x^{2}} & -\frac{2i}{\Delta t} & \left(\frac{2i}{\Delta t}-\frac{\hbar }{m\Delta x^{2}}\right) \end{array}\right)$$

The eigenvalues are $\left(0,\frac{4i}{\Delta t},-\frac{2\hbar }{m\Delta x^{2}},\frac{4i}{\Delta t}-\frac{2\hbar }{m\Delta x^{2}}\right)$ with eigenvectors (1,1,1,1), (−1,1,−1,1), (−1,−1,1,1), and (1,−1,−1,1), respectively. This is a Hadamard structure that we see repeated in **K** for the Klein–Gordon and Dirac equations [40].

We call **K** the “relations matrix” because the rows of **K** sum to zero. For example, if we were describing the line of people [Alice, Bob, Charlie, David], we would say, “Alice is in front of Bob” and “Bob is behind Alice and in front of Charlie”. Assigning a numerical value of +1 to “in front of” and −1 to “behind”, we see that adding the statements describing the locations of all four people gives a result of zero. This “summing to zero” happens because we have a self-referential, relational description of all four people. Mathematically we write $K\cdot {\left[1111\right]}^{T}=0$ and say that ${\left[1111\right]}^{T}$ is a non-null eigenvector of **K** with eigenvalue zero and **K** is said to possess a non-trivial null space. The complement to that non-trivial null space is called the “row space” of **K**. That **K** possesses a non-trivial null space is the graphical counterpart to gauge invariance and restricting the transition amplitude *Z* integral to the row space of **K** is the graphical counterpart of Fadeev–Popov gauge fixing ([39], p. 168–170). Since the source $\vec{J}$ also appears in the integral for *Z*, we need it to reside entirely in the row space of **K** which means $\vec{J}\cdot {\left[1111\right]}^{T}=0$, so that the components of $\vec{J}$ also sum to zero, i.e., $\vec{J}$ is said to be “divergence-free” which is necessary for conservation of whatever is being exchanged in the interaction. Thus, the adynamical global constraint for QM per Axiom 1 is that divergence-free $\vec{J}$ follows from relationally defined **K**, which follows from the geometric tautology ∂∂=0 encoding the construct of TTOs necessary for classical physics. As noted by Rovelli, “Gauge is ubiquitous. It is not unphysical redundancy of our mathematics. It reveals the relational structure of our world” ([26], p. 7). In addition to the action for the Schrödinger, Klein–Gordon, and Dirac equations, this relationally defined **K** appears in the Maxwell and Einstein–Hilbert graphical actions and can be extended to the graphical action for the Standard Model of particle physics [41].

#### 3.2.3. Axiom 2 and Symmetries

Axiom 2 is satisfied trivially throughout physics via coordinate invariance or general covariance, but it also has very important consequences for foundations of physics that we will articulate below. Specifically, in Section 4 we will show how conservation per Axiom 1 as restricted by Axiom 2 (NPRF) leads to two erroneous views of modern physics. First, some believe that QM entanglement means QM is incomplete ([42], p. xvii), [43] and second, some believe that QM entanglement reveals a deep rift between QM and SR ([44], p. 172), [45]. Contrary to these beliefs, we will show that QM is as complete as possible given “conservation per NPRF”, which then reveals an underlying coherence between QM and SR per the restricted Lorentz symmetry group. In general, Axiom 2 is consistent with the notion of symmetries per Hicks [46]:
There are not two worlds in one of which I am here and in the other I am three feet to the left, with everything else similarly shifted. Instead, there is just this world and two mathematical descriptions of it. The fact that those descriptions put the origin at different places does not indicate any difference between the worlds, as the origin in our mathematical description did not correspond to anything in the world anyway. The symmetries tell us what structure the world does not have.

That is, there is just one “real external world” harboring many, but always equal perspectives as far as the physics is concerned. The term “reference frame” has many meanings in physics related to microscopic and macroscopic phenomena, Galilean versus Lorentz transformations, relatively moving observers, etc. All of these ultimately represent observations of a PO (or a collection of POs) in our self-consistent collection of shared perceptual information between POs (spacetime model of the Physical). Consequently, a reference frame owes its importance to the perceptual origins (POs) of neutral monism both epistemically and ontically.

The use of symmetries to guide the progress of physics is already well established and symmetries are just another way of expressing constraints and conservation principles. The symmetry group relating non-relativistic quantum mechanics and special relativity via their mysteries as shown herein is the restricted Lorentz group. Again and again, symmetries have served to advance and unify physics, another example of principle explanation. While NPRF has profoundly counterintuitive implications such as time dilation and QM entanglement, it has not kept us from building a picture of the way nature works. On the contrary, it is what makes physics possible for us.

#### 3.2.4. The Relational Blockworld Interpretation of QM

At this point, we pause to clarify our picture of quantum-classical contextuality via our realist, psi-epistemic account of QM [41]. If one constructs the differential equation (Schrödinger equation) corresponding to the Feynman path integral, the time-dependent foliation of spacetime gives the wavefunction ψ(x,t) in concert with our time-evolved perceptions and the fact that we do not know when the outcome is going to occur. Once one has an outcome, both the configuration $x_{o}$, that is the specific spatial locations of the experimental outcomes, and time $t_{o}$ of the outcomes are fixed, so the wavefunction ψ(x,t) of configuration space becomes a probability amplitude $\psi (x_{o},t_{o})$ in spacetime, i.e., a probability amplitude for a specific outcome in spacetime. Again, the evolution of the wavefunction in configuration space before it becomes a probability amplitude in spacetime is governed by the Schrödinger equation.

However, the abrupt change from wavefunction in configuration space to probability amplitude in spacetime is not governed by the Schrödinger equation. In fact, if the Schrödinger equation is universally valid, it would simply say that the process of measurement should entangle the measurement device with the particle being measured, leaving them both to evolve according to the Schrödinger equation in a more complex configuration space (as in the relative-states formalism shown below). The Many-Worlds interpretation notwithstanding, we do not seem to experience such entangled existence in configuration space, which would contain all possible experimental outcomes. Instead, we experience a single experimental outcome in spacetime.

This contradiction between theory and experience is called the “measurement problem”. However, the time-evolved story in configuration space is not an issue with the path integral formalism as we interpret it, because we compute $\psi (x_{o},t_{o})$ directly. That is, in asking about a specific outcome we must specify the future boundary conditions that already contain definite and unique outcomes. Thus, the measurement problem is a non-starter for us. When a QM interpretation assumes the wavefunction is an epistemological tool rather than an ontological entity, that interpretation is called “psi-epistemic”.

In our path integral, constraint-based account the wavefunction in configuration space is not even used, so our account is trivially psi-epistemic. One must also fully understand the classical-quantum contextual implications of our view. The so-called “quantum system” is in fact the totality of the entire experimental set-up ([47], p. 738) such that different set-ups or configurations are not probing some autonomous quantum realm, but actually constitute different “systems”. QM then is a theory that has quantum-classical contextuality at its heart, that is its deepest lesson about the nature of physcal reality. As Ball puts it, “the quantum experiment is not probing the phenomenon but is the phenomenon” ([48], p. 90). Given our Lagrangian approach, the entire experimental set-up includes future boundary conditions: the experiment from initiation to termination in 4D spacetime, no Hilbert space required. For a detailed treatment of the measurement problem, the Born rule, and environmental decoherence see [12].

Finally, we should clarify possible sources of confusion with this quantum-classical contextuality. First, we are not saying that there exist quantum entities with inaccessible properties that are traversing the space between emitters and receivers. We are denying the existence of any worldlines for quantum momentum exchanges, no matter how large the energy or momentum being exchanged. On pain of infinite regress, a quantum exchange of momentum is not a TTO. A quantum of momentum or energy does not have any “intrinsic properties”, or intrinsic existence as we discussed in regard to neutral monism. Second, there is no quantum-classical “cut” and they do not exist independently of each other. As we pointed out, there does not seem to be any limit on the size of the quantum momentum exchange. With those emphases, we now explain how it is that some theories of physics are clearly fundamental to others even though they all satisfy our axioms.

#### 3.2.5. The Hierarchy of Physics

In order to account for the hierarchy that we know exists for theories of physics, consider Newtonian mechanics, Newtonian gravity, SR, and GR. Newtonian mechanics satisfies Axiom 1, as we just showed, and it satisfies the relativity principle per Axiom 2, but without input from Maxwell’s equations of electromagnetism and empirical evidence otherwise, Newtonian mechanics assumes the speed of light is infinite. Therefore, the Galilean transformation equations of Newtonian mechanics are simply the Lorentz transformation equations of SR with c=∞. Maxwell’s equations, which also satisfy ∂∂=0 per Axiom 1, tell us that $c=\frac{1}{\sqrt{\mu_{o}\epsilon_{o}}}$, so Axiom 2 tells us that Lorentz transformations are fundamental to Galilean transformations. Lorentz transformations are for inertial frames moving at constant velocity relative to each other and are only valid locally in the curved spacetime of GR, which allows more generally for inertial frames that are accelerating relative to each other. Newtonian gravity is then understood to result from attempting to apply a global flat frame to a curved region of spacetime, which results in the “force of gravity” where no deviation from geodesy otherwise exists, i.e., there is no “force of gravity”. Gravitational time dilation establishes GR as the more fundamental theory of gravity, so GR is fundamental to SR which is fundamental to Newtonian physics.

The bottom line is that GR is divergence-free per Axiom 1 and satisfies Axiom 2 more fundamentally than SR or Newtonian physics, since it holds for relatively accelerated reference frames. Therefore, we consider GR to be our most fundamental theory of classical physics. Assuming one insists on a classical theory that accommodates relatively accelerating inertial observers, does not involve derivatives of the metric beyond second order, and satisfies Axioms 1 and 2 (is divergence-free and gauge invariant per ∂∂=0), then GR with a cosmological constant follows necessarily. Whether or not one needs the cosmological constant Λ is an empirical question, i.e., whether $G_{\alpha \beta }=R_{\alpha \beta }-\frac{1}{2}g_{\alpha \beta }R$ or $G_{\alpha \beta }$ takes on its most general form, $G_{\alpha \beta }=R_{\alpha \beta }-\frac{1}{2}g_{\alpha \beta }R+\Lambda g_{\alpha \beta }$, is an empirically open question.

The Physical is to be modeled via quantum-classical contextuality and we have established that our most fundamental theory of classical physics is GR. That means we need our most fundamental theory of quantum physics placed in the classical context of GR. This approach to fundamental physics is not new, but it does defy conventional wisdom which has it that “gravity as we presently know it emerges from some more fundamental microscopic theory” [49]. While all our quantum theories satisfy Axiom 1 and quantum entanglement results from Axioms 1 and 2 [35,36], the most fundamental theory of quantum interactions, as of now, is the Standard Model of particle physics, so we need to incorporate this with GR without quantizing GR per se (again, gravity is not a “force” per GR). Lattice gauge theory (graphical quantum field theory) with Regge calculus (graphical GR [50]) is probably the most direct extant way to implement our version of quantum gravity, the most fundamental theory of physics possible per our quantum-classical contextuality.

#### 3.2.6. Quantum Gravity

Whether or not the three fundamental interactions of the Standard Model, i.e., SU(3) × SU(2) × U(1), are truly fundamental has yet to be determined. Certainly, the obvious algebraic extrapolation to SU(5) has been ruled out, but that is not the only pattern in the existing structure. In the lattice gauge theory formulation of the Standard Model, the bosonic gauge fields on the links of the graph supply the parallel transport for differencing fermionic vectors between nodes [40] in satisfying Axiom 1. Thus, in this view of quantum field theory, the fundamental number of vectors possible on each node is an empirically open question.

Of course, this implies that Regge calculus would supply a more fundamental classical context than GR in some regions of the spacetime manifold. For example, we expect the hot, dense early universe would not contain TTOs and might best be modeled via Regge calculus and lattice gauge theory in this approach. Does that mean we no longer have quantum-classical contextuality? No, the early universe is only part of the entire 4D spacetime manifold that also contains classical spacetime regions with stars, dust, gas, planets, telescopes, and detectors, and these TTOs then provide the necessary classical (enduring) objects for the quanta associated with the early universe.

Again, Regge calculus provides a reasonable *formalism* for the classical context of the early universe, since we might then more easily couple quantum theory to the classical context via lattice gauge theory [12]. For example, as we obtained previously [51], the Regge equation for Einstein–deSitter (EdS) cosmology with continuous time is
(11)$$\frac{\pi -\cos^{-1}\left(\frac{v^{2}/c^{2}}{2\left(v^{2}/c^{2}+2\right)}\right)-2\cos^{-1}\left(\frac{\sqrt{3v^{2}/c^{2}+4}}{2\sqrt{v^{2}/c^{2}+2}}\right)}{\sqrt{v^{2}/c^{2}+4}}=\frac{Gm}{2rc^{2}}$$

With $v^{2}/c^{2}\ll 1$ an expansion of the LHS of Equation (Equation 11) gives
(12)$$\frac{v^{2}}{4c^{2}}+O{\left(\frac{v}{c}\right)}^{4}=\frac{Gm}{2rc^{2}}$$

Thus, to leading order (defining “small” simplices required to accurately approximate GR) we have $\frac{v^{2}}{2}=\frac{Gm}{r}$, which is just a Newtonian conservation of energy expression for a unit mass moving at escape velocity *v* at distance *r* from mass *m*. In the case of Regge EdS cosmology, we see that spacetime curvature enters as a deficit angle between adjoining “Newtonian” simplices. Here you can see that the bosonic fields of the corresponding lattice gauge theory supplying parallel transport for fermionic vectors, would need to be modified per the deficit angle of the Regge simplices, the counterpart to the connection of GR.

Now, the Regge calculus solution for the scaling factor a(t) in the EdS model reproduces that of GR’s EdS solution when the spatial links are small. Spatial links are “small” when the “Newtonian” graphical velocity *v* between spatially adjacent nodes on the Regge graph is small compared to *c*, i.e., ${\left(\frac{v}{c}\right)}^{2}\ll 1$. In that case, the dynamics between adjacent spatial nodes is just Newtonian and the evolution of a(t) in Regge EdS cosmology is equal to that in GR’s EdS cosmology. When v≈2c Regge EdS cosmology encounters the “stop point problem” [52,53,54], i.e., the backward time evolution of a(t) halts, so a(t) has a minimum. Of course, this is not a real problem for Regge EdS cosmology if one is simply using it to model GR’s EdS cosmology, since one can regularly check *v* in the computational algorithm and refine the size of the lattice to ensure *v* remains small.

However, as is now obvious, we have a fundamental lattice spacing of *h* per quantum-classical reality, so further lattice refinements are not possible beyond this point and Regge calculus may deviate from GR (as would be expected in the quantum gravity regime). Thus, far from being a “problem”, the “stop point problem” actually shows us how the initial singularity is avoided in this approach to quantum gravity. This avoids the apparent inevitability of GR singularities per the Hawking–Penrose singularity theorems because the existence of timelike and null past-inextendible geodesics is entirely justified. Early models of stellar collapse used “cosmological” spheres of collapsing dust surrounded by Schwarzschild vacuum [55], so this Regge-calculus view of quantum gravity avoids the r=0 singularity of black holes precisely as it avoids the initial singularity of GR cosmology.

From these abbreviated overviews of neutral monism and physics per our axioms, it should be clear that physics, while certainly not finished, is rendered comprehensive and coherent in our view of neutral monism. In the next section, we will use this view to specifically refute two commonly held beliefs concerning QM. Therein, we will refute the inference of nonlocality from QM entanglement that some allege renders QM incompatible with SR (QM lacks coherence). And, We will refute the inference of hidden variables from QM entanglement that renders QM incomplete (QM lacks integrity).

## 4. The Axioms Reveal QM’s Completeness and Coherence

Again, some in foundations believe SR and QM are incompatible because QM entanglement seems to imply nonlocality/superluminal influences in violation of SR ([44], p. 172), [45]. Of course, we know QM is not Lorentz invariant and so it deviates trivially from SR in that fashion. As we showed above, to get QM from Lorentz invariant quantum field theory one needs to make low energy approximations. The charge of incompatibility based on QM entanglement actually carries serious consequences, because we have experimental evidence confirming the violation of Bell’s inequality per QM entanglement. If the violation of Bell’s inequality is in any way inconsistent with SR, then SR is being challenged empirically. By analogy, we know Newtonian mechanics deviates from SR because it is not Lorentz invariant. As a consequence, Newtonian mechanics predicts a very different velocity addition rule, so suppose we found experimentally that velocities do add as predicted by Newtonian mechanics. That would not merely mean that Newtonian mechanics and SR are incompatible, that would mean Newtonian mechanics has been empirically verified while SR has been empirically refuted. If one believes the violation of Bell’s inequality is in any way inconsistent with SR, and one believes the experimental evidence is accurate, then one believes SR has been empirically refuted. Clearly, that is not the case. As long as one’s account of EPR correlations is local, as ours is, there is no further worry about conflict between SR and QM.

Another belief held by some in foundations is that QM is incomplete because it accurately predicts correlations for entangled systems without spelling out any means *a la* hidden variables for why those correlations exist (Section 4.2). For example, this prompted Smolin to write ([42], p. xvii):
I hope to convince you that the conceptual problems and raging disagreements that have bedeviled quantum mechanics since its inception are unsolved and unsolvable, for the simple reason that the theory is wrong. It is highly successful, but incomplete.

Of course, this is precisely the complaint leveled by Einstein, Podolsky, and Rosen in their famous paper, “Can Quantum-Mechanical Description of Physical Reality Be Considered Complete?” [43]. As we will make clear in this section, quantum entanglement does not mean QM is wrong or incomplete. Rather, QM just needs to be understood as a principle theory, at least as it pertains to the mystery of entanglement.

What we want to do in this section is show how, per our axioms, quantum entanglement is actually evidence that QM and SR are deeply coherent and QM is complete. We begin with a reminder of the importance of reference frames in physics per our brand of neutral monism.

Again, the term “reference frame” has many meanings in physics all of which ultimately represent observations of a PO (or POs) in our self-consistent collection of shared perceptual information between POs (spacetime model of the Physical). In what follows, a measurement configuration constitutes a reference frame, as with the light postulate of SR. Therefore, again, Axiom 2 (NPRF) says that no one’s “sense experiences”, such as measurement outcomes, can evidence a privileged perspective on the “real external world” (the Physical). Let us review the consequences of the postulates of SR.

In SR, Alice is moving at velocity ${\vec{V}}_{a}$ relative to a light source and measures the speed of light from that source to be *c* ($=\frac{1}{\sqrt{\mu_{o}\epsilon_{o}}}$, as predicted by Maxwell’s equations). Bob is moving at velocity ${\vec{V}}_{b}$ relative to that same light source and measures the speed of light from that source to be *c*. Here “reference frame” refers to the relative motion of the observer and source, so all observers who share the same relative velocity with respect to the source occupy the same reference frame. NPRF in this context thus means all measurements produce the same outcome *c*, i.e., NPRF gives the light postulate.

As a consequence of this constraint we have time dilation and length contraction, which are then reconciled per NPRF via the relativity of simultaneity. That is, Alice and Bob each partition spacetime per their own equivalence relations (per their own reference frames), so that equivalence classes are their own surfaces of simultaneity. If Alice’s equivalence relation over the spacetime events yields the “true” partition of spacetime, then Bob must correct his lengths and times per length contraction and time dilation. Of course, the relativity of simultaneity says that Bob’s equivalence relation is as valid as Alice’s per NPRF. Here is an example relating time dilation, length contraction, Lorentz transformations, and the relativity of simultaneity for those who would like a reminder.

### 4.1. The Mysteries of Length Contraction and Time Dilation per Special Relativity

Suppose there are three girls moving together at 0.6c with respect to two boys. The boys and girls agree on the details of the following four events (boys’ coordinates are lower case and girls’ coordinates are upper case):Event 1: Joe meets Sara at $X_{1}=x_{1}=0$, $T_{1}=t_{1}=0$.Event 2: Bob meets Kim at $X_{2}=1250$ km, $T_{2}=-0.0025$ s, $x_{2}=1000$ km, $t_{2}=0$.Event 3: Bob meets Alice at $X_{3}=800$ km, $T_{3}=0$, $x_{3}=1000$ km, $t_{3}=0.002$ s.Event 4: Bob meets Sara at $X_{4}=0$, $T_{4}=0.00\bar{44}$ s, $x_{4}=1000$ km, $t_{4}=0.00\bar{55}$ s.

The lower-case and upper-case coordinates for each Event are related by Lorentz transformations with γ=1.25. Here is the story according to the boys.

The girls are moving in the positive *x* direction at 0.6c. Events 1 and 2 are simultaneous ($t_{1}=t_{2}=0$), so the distance between Sara and Kim is $x_{2}=1000$ km. The girls say the distance between Sara and Kim is $X_{2}=1250$ km, so that proper distance has been length contracted by γ. Event 4 happens $t_{4}=0.00\bar{55}$s after Events 1 and 2, but Sara’s clock has only ticked off $T_{4}=0.00\bar{44}$s, so her proper time has been dilated by a factor of γ. Therefore, the boys say the girls’ meter sticks are short (length contraction) and the girls’ clocks are running slow (time dilation). Here is the story according to the girls.

The boys are moving in the negative *X* direction at 0.6c. Events 1 and 3 are simultaneous ($T_{1}=T_{3}=0$), not Events 1 and 2 as the boys claim (relativity of simultaneity). Thus, the distance between Joe and Bob is $X_{3}=800$ km, not $x_{2}=1000$ km as the boys claim. Again, the proper distance has been length contracted by γ. Event 3 happens 0.0025 s after Event 2, but Bob’s clock has only ticked off $t_{3}=0.002$ s, so his proper time has been dilated by a factor of γ. Therefore, the girls say the boys’ meter sticks are short and the boys’ clocks are running slow.

In summary, Axioms 1 and 2 give the postulates of SR whence the Lorentz transformations, time dilation, length contraction, and the relativity of simultaneity. Since Alice and Bob always measure the same speed of light *c* regardless of their relative motion per NPRF, Alice says Bob’s temporal and spatial measurements need to be corrected per time dilation and length contraction while Bob says the same thing about Alice’s measurements. If NPRF is true, then neither need to be corrected (relativity of simultaneity). Thus, the mystery of SR ultimately resides in NPRF starting with the fact that everyone measures the same value for the fundamental constant *c*, i.e., the max speed of a quantum exchange of energy-momentum (Section 3). Now let us relate this mystery to the mystery of entanglement in QM as revealed by spin measurements.

### 4.2. The Mystery of Quantum Entanglement per the Bell Spin States

Bob and Alice both measure $\pm 1\left(\frac{\hbar }{2}\right)$ for all Stern–Gerlach (SG) magnet orientations when making spin measurements on a Bell spin state (Figure 3 and Figure 4). Here “reference frame” refers to the orientation of the observer’s SG magnets relative to a common reference direction (as in SR where “reference frame” refers to the observer’s motion relative to a common source). This is consistent with NPRF in that Alice and Bob are each making a direct measurement of a fundamental constant, i.e., the quantum of action *h* (Section 3), so NPRF says they must measure the same value (as with *c*). How does this lead to a mystery as in SR?

#### 4.2.1. The Mystery as Revealed by the Mermin Device

Let us borrow from Mermin’s famous example [57] of the conundrum of spin-entangled particles per the Bell spin states via the “Mermin device”. The “source” in the Mermin device (middle box in Figure 5) creates a pair of particles measured randomly by Alice and Bob in settings 1, 2, or 3 of the boxes on the left and right, respectively, in Figure 5. There are two possible outcomes of a measurement and they are denoted “R” and “G” (indicated by the light bulbs in Figure 5). Obviously, these two outcomes correspond to $\pm 1\left(\frac{\hbar }{2}\right)$ for spin-$\frac{1}{2}$ particles. Let the settings 1, 2, or 3 correspond to the three SG magnet orientations shown in Figure 6. As Mermin points out, there are two mysterious Facts about these spin measurements that Alice and Bob will discover:When Alice and Bob’s settings are the same in a given trial (“case (a)”), their outcomes are always the same, $\frac{1}{2}$ of the time RR (Alice’s outcome is R and Bob’s outcome is R) and $\frac{1}{2}$ of the time GG (Alice’s outcome is G and Bob’s outcome is G).When Alice and Bob’s settings are different in a given trial (“case (b)”), the outcomes are the same $\frac{1}{4}$ of the time, $\frac{1}{8}$ RR, and $\frac{1}{8}$ GG.

These Facts are summarized in Table 1 and represent measurements on a spin triplet state in its plane of symmetry (see below). Mermin writes:
Why do the detectors always flash the same colors when the switches are in the same positions? Since the two detectors are unconnected there is no way for one to “know” that the switch on the other is set in the same position as its own.

This leads him to introduce “instruction sets” to account for the behavior of the device when the detectors have the same settings. He writes, “It cannot be proved that there is no other way, but I challenge the reader to suggest any”. Mermin explicitly excludes the possibilities of retrocausality and superluminal communication between the particles. That is, the particles cannot “know” what settings they will encounter until they arrive at the detectors and they cannot communicate their settings and outcomes with each other in spacelike fashion. Now look at all trials when Alice’s particle has instruction set RRG and Bob’s has instruction set RRG, for example.

That means Alice and Bob’s outcomes in setting 1 will both be R, in setting 2 they will both be R, and in setting 3 they will both be G. That is, the particles will produce an RR result when Alice and Bob both choose setting 1 (referred to as “11”), an RR result when both choose setting 2 (referred to as “22”), and a GG result when both choose setting 3 (referred to as “33”). That is how instruction sets guarantee Fact 1. For different settings Alice and Bob will obtain the same outcomes when Alice chooses setting 1 and Bob chooses setting 2 (referred to as “12”), which gives an RR outcome. They will obtain the same outcomes when Alice chooses setting 2 and Bob chooses setting 1 (referred to as “21”), which also gives an RR outcome. That means we have the same outcomes for different settings in 2 of the 6 possible case (b) situations, i.e., in $\frac{1}{3}$ of case (b) trials for this instruction set. This $\frac{1}{3}$ ratio holds for any instruction set with two R(G) and one G(R).

The only other possible instruction sets are RRR or GGG where Alice and Bob’s outcomes will agree in $\frac{9}{9}$ of all trials. Thus, the “Bell inequality” for the Mermin device says that instruction sets must produce the same outcomes in more than $\frac{1}{3}$ of all case (b) trials. Indeed, if all eight instruction sets are produced with equal frequency, the RR, GG, RG, and GR outcomes for any given pair of unlike settings (12, 13, 21, 23, 31, or 32) will be produced in equal numbers, so the probability of getting the same outcomes for different settings is $\frac{1}{2}$ (Table 2). Fact 2 for QM says you only get the same outcomes in $\frac{1}{4}$ of all those trials, thereby violating the prediction per instruction sets. Thus, the conundrum of Mermin’s device, and therefore of Bell spin state measurements, is that the instruction sets needed for Fact 1 fail to yield the proper outcomes for Fact 2.

QM accurately predicts Facts 1 and 2, but does not provide any hidden variables or causal mechanism for how those Facts obtain. This leads some in foundations to believe QM is incomplete, as cited above. To understand why QM is in fact as complete as possible, we need to understand how the mysterious Facts 1 and 2 follow from Axiom 2.

#### 4.2.2. The Correlation Functions and the Bell Spin States

To understand the NPRF source of the mystery we compare the quantum and classical correlation functions. The case (a) correlation functions are the same, since instruction sets are designed that way, so we need to understand the difference for case (b). The case (b) correlation function for instruction sets from Table 2 is obviously zero, while Table 1 gives a correlation function of
(13)$$\begin{array}{c} \langle \alpha ,\beta \rangle =\left(+1\right)\left(+1\right)\left(\frac{1}{8}\right)+\left(-1\right)\left(-1\right)\left(\frac{1}{8}\right)+\left(+1\right)\left(-1\right)\left(\frac{3}{8}\right)+\left(-1\right)\left(+1\right)\left(\frac{3}{8}\right)=-\frac{1}{2} \end{array}$$

That means the Mermin device is more strongly anti-correlated for different settings than instruction sets. Indeed, QM evidences something to explain (anti-correlated results for case (b)) where instruction sets suggest there is nothing in need of explanation (uncorrelated results for case (b)). Thus, QM indicates that the conservation principle (per Axiom 1) responsible for Fact 1 of case (a) has observable implications (Fact 2) for case (b) while instruction sets say we should not expect to see any consequence of Fact 1 for case (b). While instruction sets do not predict any observable case (b) consequences for case (a) conservation, other hidden variable accounts do, but none produce the degree of correlation/anti-correlation possessed by QM. This is of course Bell’s famous result [58]. Now let us explore the nature of the conservation at work in this situation and show how it results from NPRF, i.e., how Axiom 2 constrains Axiom 1.

The four Bell spin states are
(14)$$\begin{array}{cc} |\psi_{-}\rangle = & \frac{1}{\sqrt{2}}\left(|+1-1\rangle -|-1+1\rangle \right)\\ |\psi_{+}\rangle = & \frac{1}{\sqrt{2}}\left(|+1-1\rangle +|-1+1\rangle \right)\\ |\phi_{-}\rangle = & \frac{1}{\sqrt{2}}\left(|+1+1\rangle -|-1-1\rangle \right)\\ |\phi_{+}\rangle = & \frac{1}{\sqrt{2}}\left(|+1+1\rangle +|-1-1\rangle \right) \end{array}$$
in the $\sigma_{z}$ eigenbasis, say. The eigenvalues for any 2×2 Hermitian matrix can be written +1 and –1, so whatever Alice and Bob are measuring it can be said to give outcomes of +1 or –1. $|\psi_{-}\rangle$ (the spin singlet state) is invariant under all three SU(2) transformations $e^{i\Theta \sigma_{j}}$, where $\sigma_{j}$ are the Pauli spin matrices with j={x,y,z} and $\Theta =\frac{\theta }{2}$ is the angle in Hilbert space [35]. Thus, $|\psi_{-}\rangle$ says that when the SG magnets are aligned (Alice and Bob are in the same reference frame) the outcomes are always opposite (+− or −+). Since $|\psi_{-}\rangle$ has that same functional form under any SU(2) transformation in Hilbert space representing SO(3) rotation in any plane of real space, the outcomes are always opposite (+− and −+) for aligned SG magnets in any plane. That is the “SO(3) conservation” (per Axiom 1) associated with this SU(2) symmetry. The invariance of $|\psi_{-}\rangle$ under all three SU(2) transformations makes sense, since the spin singlet state represents the conservation of a total spin angular momentum of S=0, which is directionless, and each SU(2) transformation in Hilbert space corresponds to an element of SO(3) in real space. Now for the spin triplet states.

The first spin triplet shown $|\psi_{+}\rangle$ is invariant under $e^{i\Theta \sigma_{z}}$, the second $|\phi_{-}\rangle$ is invariant under $e^{i\Theta \sigma_{x}}$, and the third $|\phi_{+}\rangle$ is invariant under $e^{i\Theta \sigma_{y}}$[35]. The invariance of each of the spin triplet states under its respective SU(2) transformation in Hilbert space represents the conserved spin angular momentum S=1 (in units of ℏ=1 for spin-$\frac{1}{2}$ particles) for each of the planes xz ($|\phi_{+}\rangle$), yz ($|\phi_{-}\rangle$), and xy ($|\psi_{+}\rangle$) in real space. (The state $|\psi_{+}\rangle$ is $i|\phi_{+}\rangle$ in the $\sigma_{y}$ eigenbasis and $-|\phi_{-}\rangle$ in the $\sigma_{x}$ eigenbasis, so it gives like results for like settings in the xy-plane of symmetry.) Specifically, when the SG magnets are aligned (the measurements are being made in the same reference frame) anywhere in the respective plane of symmetry the outcomes are always the same. It is a planar conservation according to our analysis and our experiment would determine which plane (see [59] for a spin-1 example). The binary outcomes represent bi-directionality in the plane of symmetry as with spin-$\frac{1}{2}$ particles, or they represent axial duality perpendicular to the plane of symmetry as with spin-1 particles [36]. Now we compute the correlation functions.

If Alice is making her spin measurement $\sigma_{1}$ in the $\hat{a}$ direction and Bob is making his spin measurement $\sigma_{2}$ in the $\hat{b}$ direction, we have
(15)$$\begin{array}{cc} \; & \sigma_{1}=\hat{a}\cdot \vec{\sigma }=a_{x}\sigma_{x}+a_{y}\sigma_{y}+a_{z}\sigma_{z}\\ \; & \sigma_{2}=\hat{b}\cdot \vec{\sigma }=b_{x}\sigma_{x}+b_{y}\sigma_{y}+b_{z}\sigma_{z} \end{array}$$

The correlation functions are given by
(16)$$\begin{array}{ccc} \; & \langle \psi_{-}|\sigma_{1}\sigma_{2}|\psi_{-}\rangle = & -a_{x}b_{x}-a_{y}b_{y}-a_{z}b_{z}\\ \; & \langle \psi_{+}|\sigma_{1}\sigma_{2}|\psi_{+}\rangle = & a_{x}b_{x}+a_{y}b_{y}-a_{z}b_{z}\\ \; & \langle \phi_{-}|\sigma_{1}\sigma_{2}|\phi_{-}\rangle = & -a_{x}b_{x}+a_{y}b_{y}+a_{z}b_{z}\\ \; & \langle \phi_{+}|\sigma_{1}\sigma_{2}|\phi_{+}\rangle = & a_{x}b_{x}-a_{y}b_{y}+a_{z}b_{z} \end{array}$$

That is, the quantum correlation function is given by $-\cos\theta =-\hat{a}\cdot \hat{b}$ in any plane for the spin singlet state and $\cos\theta =\hat{a}\cdot \hat{b}$ in the plane of symmetry for the spin triplet states. Thus, the correlation function for any pair of case (b) settings in the Mermin device (Figure 6) is $\cos\left(120^{\circ }\right)=-\frac{1}{2}$, in agreement with Equation (Equation 13), instead of zero per that of instruction sets. In other words, the Mermin device represents spin measurements on an S=1 spin-entangled pair of particles in their plane of symmetry at the angles given by Figure 6. If you let Bob’s R(G) results represent Alice’s G(R) results, the Mermin device then represents spin measurements on the spin singlet state in some plane (all planes are planes of symmetry for S=0). In that case, the correlation function for any pair of case (b) settings in the Mermin device is $-\cos\left(120^{\circ }\right)=\frac{1}{2}$, instead of zero per that of instruction sets. For the S=0 case (b) situation, the Mermin device is giving us correlated results rather than uncorrelated results per instruction sets. For the S=1 case (b) situation, the Mermin device is giving us anti-correlated results rather than uncorrelated results per instruction sets. Why is that and how is that related to NPRF?

#### 4.2.3. Axiom 2 and the Quantum Correlation Function

Assuming only that Alice and Bob measure ±1 with equal frequency at some settings α and β, respectively, the correlation function is [35,36]
(17)$$\langle \alpha ,\beta \rangle =\frac{1}{2}{\left(+1\right)}_{A}\bar{BA+}+\frac{1}{2}{\left(-1\right)}_{A}\bar{BA-}$$
where $\bar{BA+}$ is what Bob averaged when Alice measured +1 (denoted ${\left(+1\right)}_{A}$) and $\bar{BA-}$ is what Bob averaged when Alice measured −1 (denoted ${\left(-1\right)}_{A}$). In other words, we have partitioned the data per Alice’s equivalence relation, i.e., Alice’s +1 results and Alice’s −1 results. Note that this correlation function is independent of the formalism of QM. Let us continue to analyze the situation from Alice’s perspective.

We will review the spin triplet state case, since the spin singlet state is analogous [35] (Figure 7 and Figure 8). In classical physics, one would say the projection of the spin angular momentum vector of Alice’s particle ${\vec{S}}_{A}=+1\hat{\alpha }$ along $\hat{\beta }$ is ${\vec{S}}_{A}\cdot \hat{\beta }=+\cos\left(\theta \right)$ where again θ is the angle between the unit vectors $\hat{\alpha }$ (= $\hat{a}$) and $\hat{\beta }$ (= $\hat{b}$). That is because the prediction from classical physics is that all values between $+1\left(\frac{\hbar }{2}\right)$ and $-1\left(\frac{\hbar }{2}\right)$ are possible outcomes for a spin measurement (Figure 3). From Alice’s perspective, had Bob measured at the same angle, i.e., β=α, he would have found the spin angular momentum vector of his particle was ${\vec{S}}_{B}={\vec{S}}_{A}=+1\hat{\alpha }$, so that ${\vec{S}}_{A}+{\vec{S}}_{B}={\vec{S}}_{Total}=2$ (this is S = 1 in units of $\frac{\hbar }{2}=1$). Since he did not measure the spin angular momentum of his particle at the same angle, he should have obtained a fraction of the length of ${\vec{S}}_{B}$, i.e., ${\vec{S}}_{B}\cdot \hat{\beta }=+1\hat{\alpha }\cdot \hat{\beta }=\cos\left(\theta \right)$ (Figure 9). Of course, Bob only ever obtains +1 or −1, but suppose Bob’s results *average*cos(θ) (Figure 8 and Figure 10), that is
(18)$$\bar{BA+}=\cos\left(\theta \right)$$
and similarly
(19)$$\bar{BA-}=-\cos\left(\theta \right)$$

Then, putting these into Equation (Equation 17) we obtain
(20)$$\langle \alpha ,\beta \rangle =\frac{1}{2}{\left(+1\right)}_{A}\left(\cos\left(\theta \right)\right)+\frac{1}{2}{\left(-1\right)}_{A}\left(-\cos\left(\theta \right)\right)=\cos\left(\theta \right)$$
which is the same as the quantum correlation function for the planar S=1 conservation of spin angular momentum shown above. Notice that Equations (Equation 18) and (Equation 19) are mathematical facts for obtaining the quantum correlation function, we are simply motivating these facts via conservation of spin angular momentum in accord with the symmetries shown above. Of course, Bob could partition the data according to his equivalence relation (per his reference frame) and claim that it is Alice who must average her results (obtained in her reference frame) to conserve spin angular momentum.

#### 4.2.4. Completing the Analogy with SR per Axiom 2

Now we can complete our analogy with SR above. In order to satisfy conservation of spin angular momentum per Axiom 1 for any given trial when Alice and Bob are making different measurements, i.e., when they are in different reference frames, it would be necessary for Bob or Alice to measure some fraction, ±cos(θ). For example, if Alice measured +1 at α=0 for an S=1 state (in the plane of symmetry) and Bob made his measurement (in the plane of symmetry) at $\beta =60^{\circ }$, then Bob’s outcome would need to be $\frac{1}{2}$ (Figure 10). In that case, we would know that Alice measured the “true” spin angular momentum of her particle while Bob only measured a component of the “true” spin angular momentum for his particle. Thus, Alice’s SG magnet orientation would definitely constitute a “preferred reference frame” in violation of Axiom 2.

This is precisely what does *not* happen. Alice and Bob both always measure $\pm 1\left(\frac{\hbar }{2}\right)$, no fractions, in accord with NPRF. This fact alone distinguishes the quantum joint distribution from the classical joint distribution for the Bell spin states [56] (Figure 3). Therefore, the “average-only” conservation responsible for the correlation function for the Bell spin states leading to Facts 1 and 2 for the Mermin device is actually conservation per Axiom 1 as restricted by Axiom 2 (NPRF).

In other words, Alice and Bob each partition the data per their own equivalence relations (per their own reference frames), so that equivalence classes are their own +1 and −1 data events. If Alice’s equivalence relation over the data events yields the “true” partition of the data, then Bob must correct (average) his results per “average-only” conservation. Of course, NPRF says that Bob’s equivalence relation is as valid as Alice’s, which we might call the “relativity of data partition” (Figure 11).

Thus, quantum entanglement does not imply that QM is incomplete. The “average-only” conservation that leads some to infer that QM is incomplete follows from Axiom 1 as restricted by Axiom 2 (NPRF). And, QM entanglement does not imply that QM and SR are incompatible. The mysteries of SR (time dilation and length contraction) ultimately obtain for the same reason as the mystery of Bell state entanglement in QM, i.e., the restriction on Axiom 1 by Axiom 2, further relating the fundamental constants *c* and *h*, respectively. Notice that for QM and SR, we have a principle explanation of the phenomena using adynamical global constraints and this leads to mystery per our desire for constructive explanation in accord with our dynamical bias. At least a good deal of both QM and relativity can be seen as theories about adynamical global constraints on what POs can experience, observe, measure, etc. Quantum entanglement does not render QM and SR incompatible, but quite the opposite, it reveals a deep coherence between them. Keep in mind that per our neutral monism this unifying principle explanation is both psychology and physics. In the next section we want to apply this sort of adynamical global constraint thinking to the Hardy-type case we mentioned at the beginning of the paper.

## 5. QM and Experience

We are all aware that there are many cases where the rules of QM (at least on certain interpretations) conflict with the rules that supposedly govern our everyday classical experiences. For example, POs always experience definite outcomes even when QM tells us maybe we should not. Of course on our view as outlined above, that simple cannot happen. However, to dispel such puzzles and mysteries one has to let go of the dynamical/constructive bias that leads inexorably to these problems. In this regard let us discuss the Hardy-type case where there is a possible tension between how we (POs) experience the world and some QM experimental set-up, in this case we are talking about delayed choice quantum eraser. In this section, we consider constraint-based explanation for the delayed choice quantum eraser experiment. In order to bring this possible tension out most fully we will alter the set-up of the experiment by adding a conscious agent (PO) who attempts to violate the probabilities of QM, as one might think a truly free conscious agent ought to be able to do. Let us start with a description of the experiment.

Using pictures from Hillmer and Kwiat [60] we start with a particle interference pattern (Figure 12) then we scatter photons off the particles after they have passed through the slits(s) (Figure 13) and finally we erase the which-way information obtained by the scattered photons by inserting a lens (Figure 14).

In the Hillmer and Kwiat article the lens (eraser) is inserted after the particles have passed through the slits, but experiments have been done where the “lens is inserted” after the particles have hit the detector. This is called a “delayed choice quantum eraser experiment” [61]. The question from our dynamical perspective is, How do the particles “know” whether or not the lens will be inserted? And, if they do not “know” whether the lens will be inserted or not, how do they “know” whether or not to create the interference pattern? These questions assume temporally sequential, causal explanation, i.e., we are assuming dynamical/constructive explanation.

If we rather seek an adynamical, 4D constraint-based/principle explanation, we are content with the fact per QM that the distribution of particles on the screen is consistent with the presence or absence of the lens in spacetime. The insertion of the lens does not “cause” the interference pattern any more than the interference pattern “causes” the insertion of the lens. No new physics is needed to explain this phenomenon, just the willingness to rise to Wilczek’s challenge ([62], p. 37):
A recurring theme in natural philosophy is the tension between the God’s-eye view of reality comprehended as a whole and the ant’s-eye view of human consciousness, which senses a succession of events in time. Since the days of Isaac Newton, the ant’s-eye view has dominated fundamental physics. We divide our description of the world into dynamical laws that, paradoxically, exist outside of time according to some, and initial conditions on which those laws act. The dynamical laws do not determine which initial conditions describe reality. That division has been enormously useful and successful pragmatically, but it leaves us far short of a full scientific account of the world as we know it. The account it gives—things are what they are because they were what they were—raises the question, Why were things that way and not any other? The God’s-eye view seems, in the light of relativity theory, to be far more natural. ...*To me, ascending from the ant’s-eye view to the God’s-eye view of physical reality is the most profound challenge for fundamental physics in the next 100 years* [italics ours].

Let us now bring the conscious agent (PO) into the picture by imagining it is a conscious agent inserting the lens (or not) in the experimental set-up. The question from our dynamical perspective is, What will the PO experience if he or she is the agent deciding whether or not to insert the lens? If the predictions of QM are to hold, then their decision must always be in accord with the particle’s behavior at the detection screen and that event occurred before the PO made the decision. Assuming QM holds, will the PO feel mentally “coerced” into making the appropriate choice? Will he or she feel some “physical force” moving their hand against their will? Most people do not like the idea that our “freely made” decisions can be the result of a single particle striking a distant detector. It would seem that QM does not care about choice at all, delayed or otherwise.

While most people predict that a conscious agent will not violate the probabilities of QM anymore than a classical measuring device, Hardy has proposed an experiment to test this fact. Concerning such an experimental test, he states [7]:
[If] you only saw a violation of quantum theory when you had systems that might be regarded as conscious, humans or other animals, that would certainly be exciting. I can’t imagine a more striking experimental result in physics than that. We’d want to debate as to what that meant. It wouldn’t settle the question, but it would certainly have a strong bearing on the issue of free will.

What explains the agreement between the agent’s decision and the particle’s pattern if it is not “spooky action at a distance” or “backwards causation?” Why does the conscious agent always make the “right” choice in accord with QM? One doubts there is some special new physical or mental force acting on the hand or mind of the conscious observer. For us the answer is simple—we instead ignore our anthropocentric bias and allow for the possibility that objective reality is fundamentally the 4D universe whose various patterns/distributions are determined fundamentally by adynamical global 4D-constraints, not by dynamical laws/processes acting on matter and conscious mind conceived a la de facto dualism to make it move/decide. We can then accept that there are some constraint-based explanations that do not allow for dynamical counterparts, at least dynamical counterparts without serious baggage, such as those discussed earlier. The constraint-based explanation here is the distribution of quantum energy-momentum exchanges in the spacetime context for the experimental set-up and procedure, as explained in Section 3.

The point is, adynamical global constraints in spacetime also constrain, given neutral monism trivially so, the conscious choices of conscious agents and their experiences. Thus, physics a la neutral monism is already part of psychology in that it places real constraints on what can be experienced to include memories (classical records) and choices. Conscious agents attempting to override QM do not experience any weird forces acting on them because there are no such forces. It is simply the case that the POs conscious choices will be made in accord with the relevant adynamical global constraints per the reality of the past, present, and future. Such agents feel like they have Libertarian free will (that the future is open) because they experience reality from the “ant’s-eye” view. For a lengthy formal application of our way of thinking to recent worries about Wigner’s friend, which has led some people to claim that QM is not consistent with an objective reality (i.e., one in which measurement outcomes are observer-independent and there will be no in-principle disagreements between observers), see [63]. The point is, given our brand of neutral monism, the axiomatic constraints of psychology-physics upon spacetime and POs guarantees there is an objective reality, at least as long as those constraints are in play.

## 6. Conclusions

We understand that some readers might find our approach “too radical” because it violates several core intuitions:Mental/physical property dualism as found in both strong emergence and panpsychism.A Newtonian as opposed to relativistic conception of time as it pertains to “physical” and conscious “mental” processes.Ontological and methodological reductionism. The idea that physics is fundamental and that in principle, the smallest scale phenomena always explain the larger scale phenomena, but never the reverse. The foundationalism or hierarchy thesis wherein relations between the smallest scale physical entities and other larger scale physical entities are anti-symmetric, transitive, and anti-reflexive.The fundamentality of causal and dynamical explanation over principle-type explanation.

Taken together, these assumptions generally lead people to believe that the hard problem and explanatory gap are unavoidable mysteries wherein we are forced to start with quantum field theory, GR, or quantum gravity as fundamental and then attempt to dynamically or causally explain phenomenal consciousness from there. Thus we are forced into one of two deeply flawed positions, strong emergence or panpsychism. These assumptions also create major roadblocks for explaining quantum “weirdness” and unifying physics. Maybe it is time to go back and start questioning these assumptions. This is why we do not think our approach is “too radical”. Furthermore, once we free ourselves of Galileo’s primary/secondary property distinction in favor of neutral monism, we see immediately that physics has always and integrally been about conscious experience. Just as there is no metaphysical gap between matter and mind in neutral monism, there is a union between physics and psychology.

## Figures and Tables

**Figure 1 entropy-22-00551-f001:**
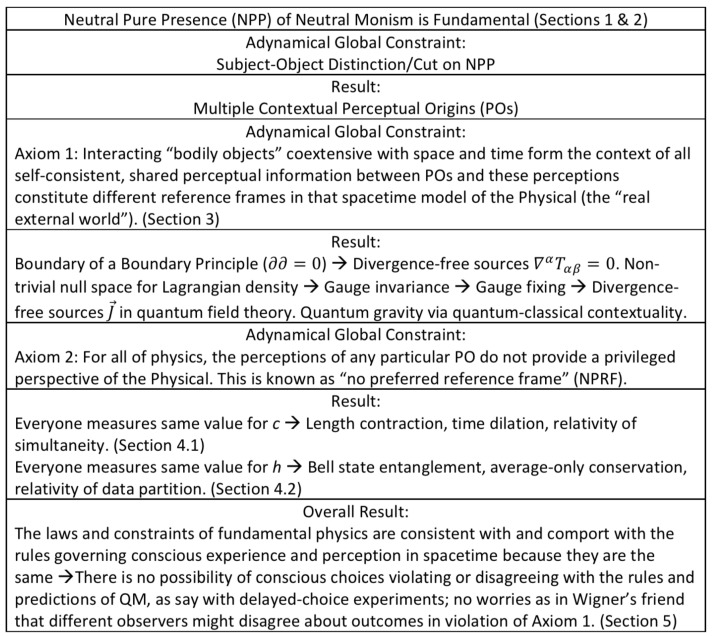
**Physics From Neutral Monism**. This is a summary of the paper. Note that what is being described is not a causal, dynamical, or temporal process but a number of adynamical global constraints that operate over spacetime understood per neutral monism. Thus “events” in Russell’s sense do not emerge in some dynamical, causal, or temporal sense from Neutral Pure Presence. Yet, the world of experience is grounded in and one with Neutral Pure Presence. The experience of temporal flow, as well as causal and dynamical processes, are strictly sensible only “within” spacetime with POs (the subject-object cut). The remaining adynamical global constraints ensure that POs experience a world of interacting trans-temporal objects, a world where one frame of reference or coordinate system can always be related to another with no inconsistency. One must always keep in mind that this is neutral monism wherein subjectivity and the objective world are fundamentally one—the difference is perspectival only. Spacetime (the world of experience) is not a noumenal physical arena or virtual construction of the mind, in which there are subjects or selves conceived dualistically, with qualia in their heads. The mind or self is not “in the head” or brain, it is extended in spacetime, part of the world of experience, and interacts directly with that world.

**Figure 2 entropy-22-00551-f002:**
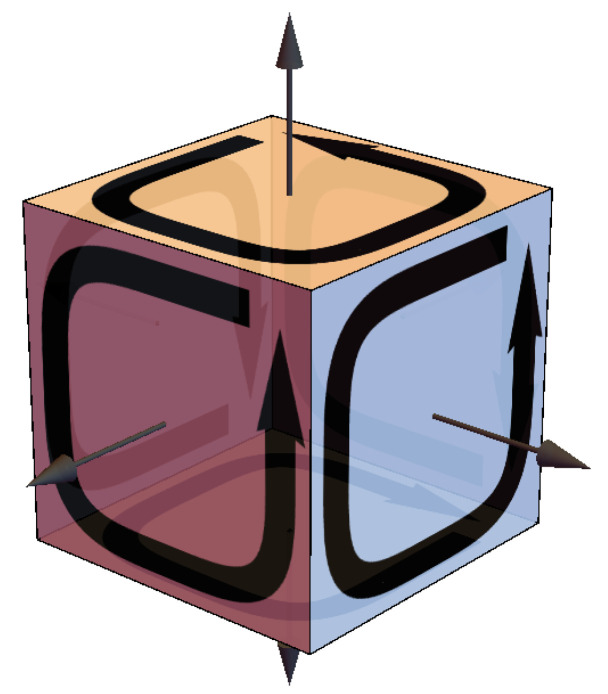
The boundary of a boundary is zero. The oriented plaquettes bound the cube and the directed edges bound the plaquettes. As you can see from the picture, every edge has oppositely oriented directions that cancel out. Thus, the boundaries of the plaguettes (the edges), which bound the cube, sum to zero.

**Figure 3 entropy-22-00551-f003:**
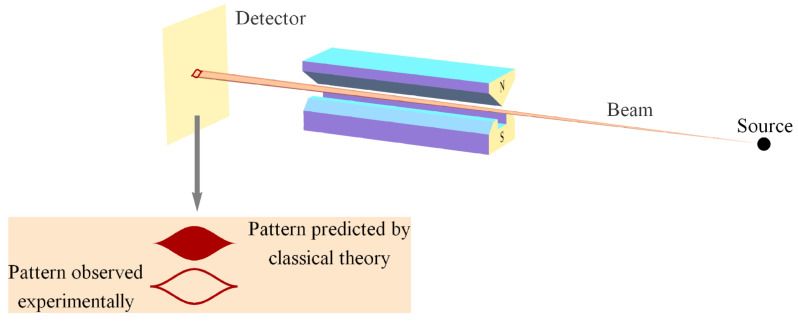
A Stern–Gerlach (SG) spin measurement showing the two possible outcomes, up and down, represented numerically by +1 and −1, respectively. The important point to note here is that the classical analysis predicts all possible deflections, not just the two that are observed. This difference uniquely distinguishes the quantum joint distribution from the classical joint distribution for the Bell spin states [56].

**Figure 4 entropy-22-00551-f004:**
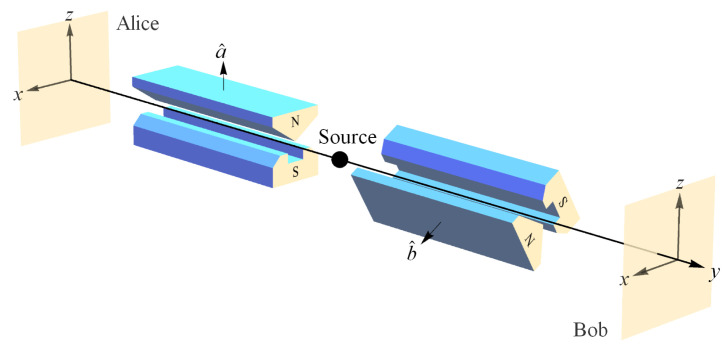
Alice and Bob making spin measurements on a pair of spin-entangled particles with their Stern–Gerlach (SG) magnets and detectors in the xz-plane.

**Figure 5 entropy-22-00551-f005:**
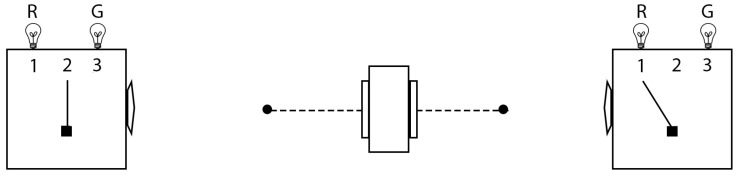
**The Mermin Device**. Alice has her measuring device on the left set to 2 and Bob has his measuring device on the right set to 1. The particles have been emitted by the source in the middle and are in route to the measuring devices.

**Figure 6 entropy-22-00551-f006:**
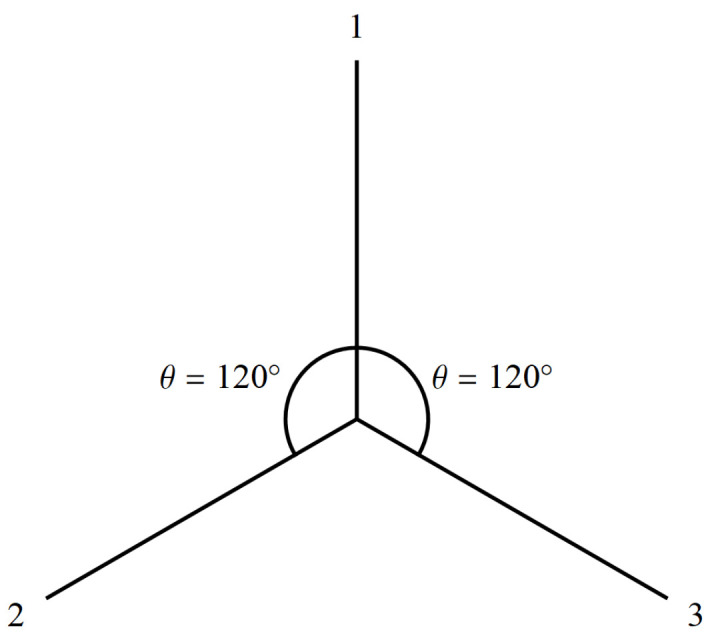
**Three possible orientations** of Alice and Bob’s SG magnets for the Mermin device.

**Figure 7 entropy-22-00551-f007:**
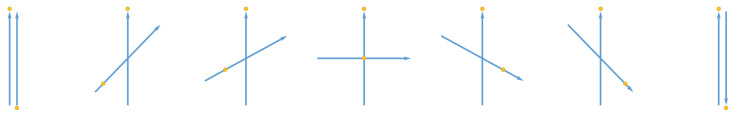
**Average View for the Spin Singlet State**. Reading from left to right, as Bob rotates his SG magnets relative to Alice’s SG magnets for her +1 outcome, the average value of his outcome varies from −1 (totally down, arrow bottom) to 0 to +1 (totally up, arrow tip). This obtains per conservation of spin angular momentum on average in accord with no preferred reference frame. Bob can say exactly the same about Alice’s outcomes as she rotates her SG magnets relative to his SG magnets for his +1 outcome. That is, their outcomes can only satisfy conservation of spin angular momentum on average in different reference frames, because they only measure ±1, never a fractional result. Thus, just as with the light postulate of SR, we see that no preferred reference frame requires quantum outcomes $\pm 1\left(\frac{\hbar }{2}\right)$ for all measurements leading to constraint-based explanation.

**Figure 8 entropy-22-00551-f008:**
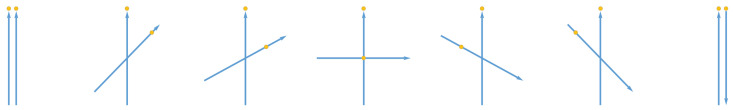
**Average View for the Spin Triplet States**. Reading from left to right, as Bob rotates his SG magnets relative to Alice’s SG magnets for her +1 outcome, the average value of his outcome varies from +1 (totally up, arrow tip) to 0 to −1 (totally down, arrow bottom). This obtains per conservation of spin angular momentum on average in accord with no preferred reference frame. Bob can say exactly the same about Alice’s outcomes as she rotates her SG magnets relative to his SG magnets for his +1 outcome. That is, their outcomes can only satisfy conservation of spin angular momentum on average in different reference frames, because they only measure ±1, never a fractional result. Thus, just as with the light postulate of SR, we see that no preferred reference frame requires quantum outcomes $\pm 1\left(\frac{\hbar }{2}\right)$ for all measurements leading to constraint-based explanation.

**Figure 9 entropy-22-00551-f009:**
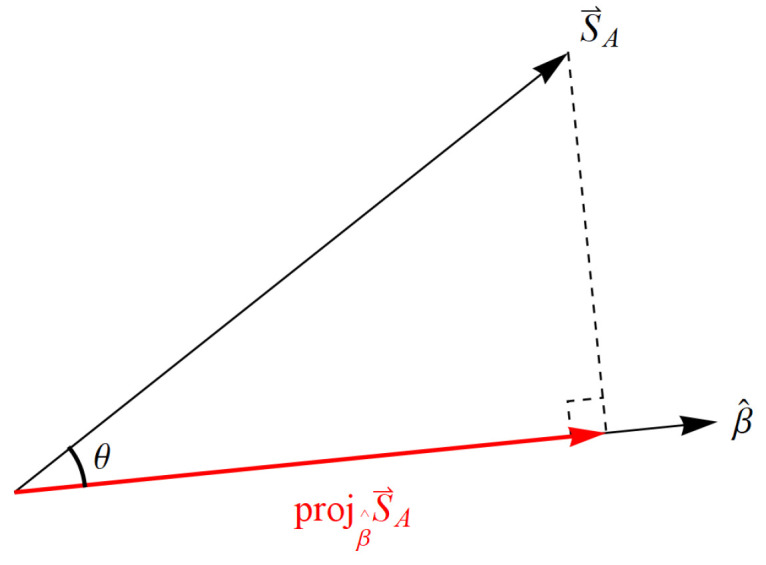
The spin angular momentum of Bob’s particle ${\vec{S}}_{B}={\vec{S}}_{A}$ projected along his measurement direction $\hat{\beta }$. This picture violates Axiom 2 (NPRF) and does not produce the QM outcomes.

**Figure 10 entropy-22-00551-f010:**
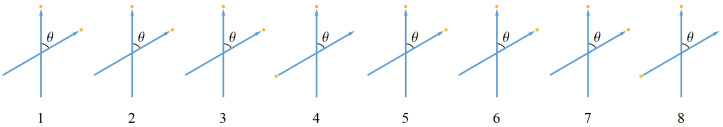
A spatiotemporal ensemble of 8 experimental trials for the spin triplet states showing Bob’s outcomes corresponding to Alice’s +1 outcomes when $\theta =60^{\circ }$. Spin angular momentum is not conserved in any given trial, because there are two different measurements being made, i.e., outcomes are in two different reference frames, but it is conserved on average for all 8 trials (six up outcomes and two down outcomes average to $\cos60^{\circ }=\frac{1}{2}$). It is impossible for spin angular momentum to be conserved explicitly in each trial since the measurement outcomes are binary (quantum) with values of +1 (up) or −1 (down) per no preferred reference frame. The “SO(3) conservation” at work here does not assume Alice and Bob’s measured values of spin angular momentum are mere components of some hidden angular momentum (Figure 9). That is, the measured values of spin angular momentum *are* the angular momenta contributing to this “SO(3) conservation.”

**Figure 11 entropy-22-00551-f011:**
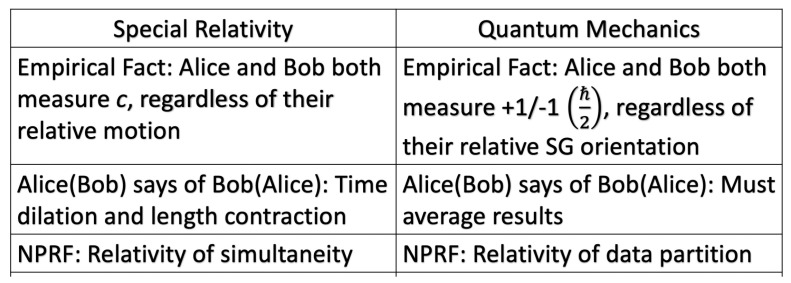
**Comparing special relativity with quantum mechanics according to no preferred reference frame (Axiom 2)**. Alice and Bob both measure the same speed of light *c* regardless of their relative motion per NPRF, therefore Alice(Bob) may claim that Bob’s(Alice’s) length and time measurements are erroneous and need to be corrected (length contraction and time dilation). Likewise, because Alice and Bob both measure the same values for spin angular momentum ±1$\left(\frac{\hbar }{2}\right)$ regardless of their relative SG magnet orientation per NPRF, Alice(Bob) may claim that Bob’s(Alice’s) individual ±1 values are erroneous and need to be corrected (averaged, Figure 7, Figure 8, Figure 9 and Figure 10). In both cases, NPRF resolves the mystery it creates. In SR, the apparently inconsistent results can be reconciled via the relativity of simultaneity. That is, Alice and Bob each partition spacetime per their own equivalence relations (per their own reference frames), so that equivalence classes are their own surfaces of simultaneity and these partitions are equally valid per NPRF. This is completely analogous to QM, where the apparently inconsistent results per the Bell spin states arising because of NPRF can be reconciled by NPRF via the “relativity of data partition”. That is, Alice and Bob each partition the data per their own equivalence relations (per their own reference frames), so that equivalence classes are their own +1 and −1 data events and these partitions are equally valid.

**Figure 12 entropy-22-00551-f012:**
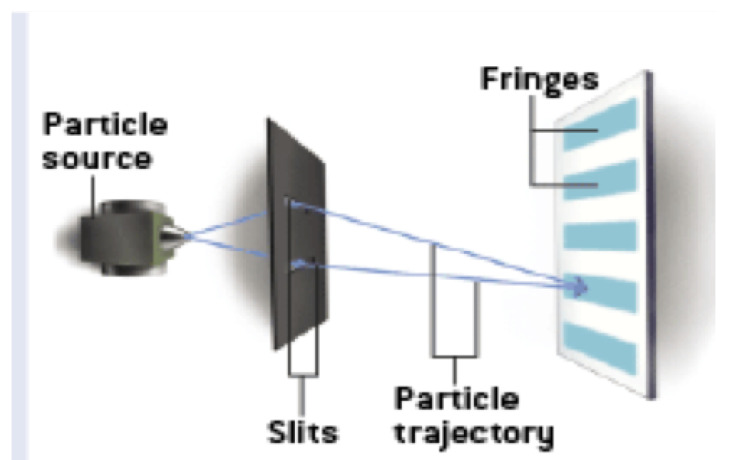
Particles create an interference pattern when proceeding through the double slits.

**Figure 13 entropy-22-00551-f013:**
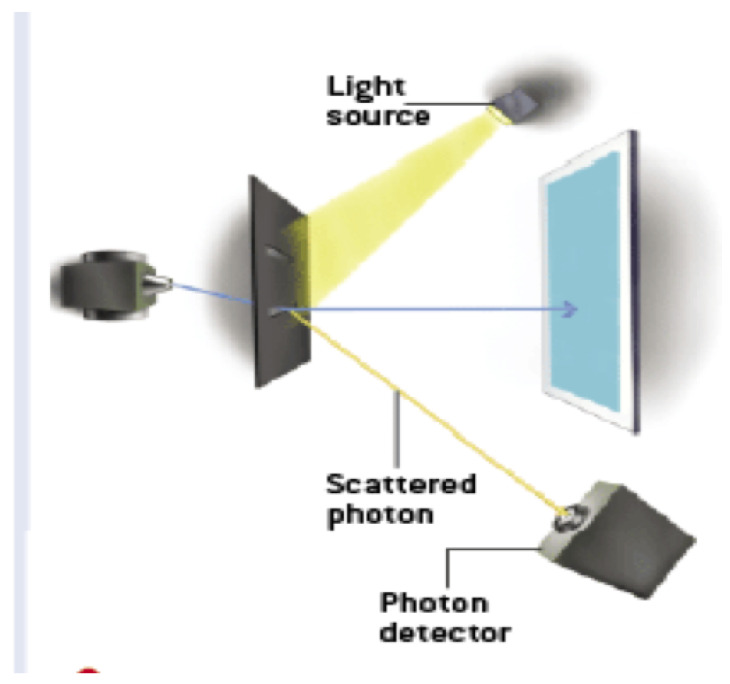
The interference pattern of Figure 12 can be destroyed by scattering photons and using those scattered photons to determine which slit the particle went through on each trial.

**Figure 14 entropy-22-00551-f014:**
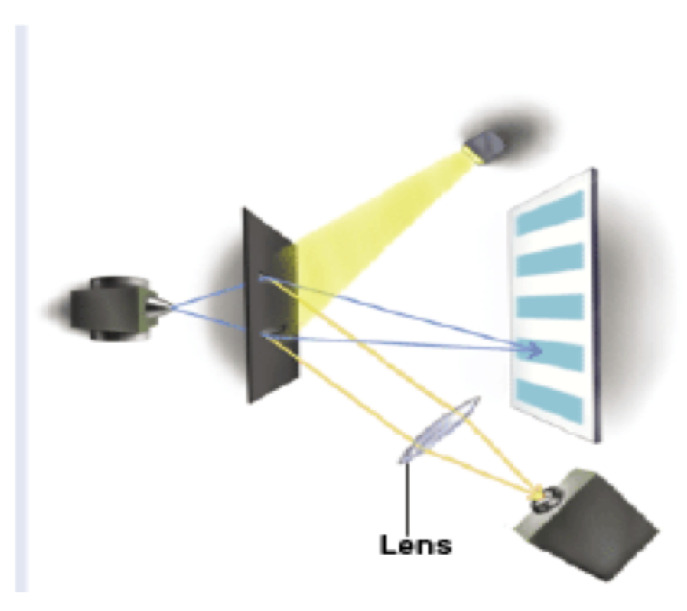
The interference pattern of Figure 12 can be restored after scattering photons as in Figure 13 by destroying the which-way information in the scattered photons (here done by inserting a lens). This is known as “quantum eraser.”

**Table 1 entropy-22-00551-t001:** Summary of outcome probabilities for the Mermin device.

Case (a) Same Settings				Case (b) Different Settings			
		Alice				Alice	
		R	G			R	G
Bob	R	1/2	0	Bob	R	1/8	3/8
	G	0	1/2		G	3/8	1/8

**Table 2 entropy-22-00551-t002:** Summary of outcome probabilities for instruction sets. We are assuming the eight possible instruction sets are produced with equal frequency.

Case (a) Same Settings				Case (b) Different Settings			
		Alice				Alice	
		R	G			R	G
Bob	R	1/2	0	Bob	R	1/4	1/4
	G	0	1/2		G	1/4	1/4

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
