# Peer review of "Re-Thinking the World with Neutral Monism:Removing the Boundaries Between Mind, Matter, and Spacetime"

_entropy, 2020, doi:10.3390/e22050551_

Round 1

Reviewer 1 Report

This article is pointing in the proper direction, relating consciousness through neutral monism to fundamental physics and the nature of spacetime. This necessitates dealing with quantum mechanics, measurement, entanglement, special and general relativity, loop quantum gravity and spin networks. This is quite an ambitious agenda, but I agree it is the proper approach.

In the Introduction the authors hope “the Galileo or Galileos of consciousness studies will figure out ways to formally model and explain various aspects of conscious processes, and maybe even relate ….these to neural processes or more fundamental physics.”

Indeed, I agree. But the authors (mysteriously, to me) omit mention of the work of Roger Penrose who might very well already be the “Galileo of consciousness studies”. Penrose is the elephant in the room, providing a potential resolution of precisely what the authors are looking for, and yet, incredulously, he’s not even mentioned. There are no references to his work, or to the Orch OR theory of consciousness in the brain he co-developed with Stuart Hameroff. This may be due to an aversion to quantum state reduction, collapse of the wavefunction, but in an article the authors feel may be “too radical”, they balk at the most fulsome argument available to reconcile their hypothesis. If Penrose is too radical, then this article is “not radical enough”. If the authors disagree, or find fault with the Penrose approach, they should explain why. Ignoring the most relevant, pioneering work in the field is a glaring omission.

Neutral monism is based on the idea that both matter and mind derive from one underlying entity which, by itself, is neither material not mental. The measurement problem is the issue that quantum superpositions are known to exist, but cannot be measured or observed. Some believe conscious observation causes quantum state reduction, collapse of the wavefunction, but this puts consciousness outside science, a dualist position, which the authors reject. Its not clear to me the authors’ favored take on measurement – is it multiple words?

In 1989 Penrose wrote ‘The emperor’s new mind’ which suggested that 1) superpositions were separated curvatures in underlying spacetime geometry (connecting general relativity and quantum physics, a stated goal of the authors) 2) superpositions/separated spacetime curvatures are unstable and their wavefunctions spontaneously self-collapse, undergo quantum state reduction due to an objective threshold, ‘Objective reduction’, ‘OR’, 2) that such OR events occurred at time t = h/E (where h is the Planck Dirac constant, and E the gravitational self energy of the superposition, and 3) each such moment results in a moment of (proto-conscious) phenomenal experience, a quale. To implement OR in the brain, and organize proto-conscious moments into full, rich consciousness, Penrose teamed with Hameroff for the ‘orchestrated objective reduction’ ‘Orch OR’ theory based on orchestrated quantum computations in brain microtubules which terminate/halt by  

OR fits perfectly in the framework of neutral monism.

                                                   Superpositioned/

                                                separated spacetime

                                                         |                   |

                                                      |         OR           |

                                                   |                                 |

                                            Matter                          Mental

Putting aside the brain, microtubules and Orch OR, Penrose OR basically proposes that a process in underlying (neutral) spacetime geometry undergoes OR to produce 1) a specific state of matter, and 2) a moment of subjective experience.   OR is a neutral monist’s dream.

The authors discuss loop quantum gravity, spin networks, QM and consciousness. Penrose is a pioneer in all these areas. He gave a talk at the Oxford conference which is the context of this paper. Why is his work not mentioned?

Some random comments:

page 3 

“One might think emergent panpsychism is true and that QM entanglement conceived as fusion answers for the combination problem” (ref 5, Seager)

Entanglement in the brain may indeed solve the combination problem (though panpsychism has other problems) and the binding problem in cognition and consciousness. But it would require quantum brain biology, and the best scientific approach to brain-wide quantum entanglement at warm brain temperatures is the Penrose-Hameroff Orch OR theory (entanglement among quantum states within tubulins in microtubules). How else would you avoid decoherence? How can one blithely invoke biological quantum entanglement without acknowledging the work that went into making quantum biology plausible?

“we want to know what is the most fundamental and universal ontological and formal relationship between conscious experience and relativity and QM….”

Again, Penrose OR is precisely what you need.

Page 4 (citing James and Russell)

“ what we call spacetime is nothing but the events therein,…..”

 First, I would include Whitehead and his process philosophy, “occasions of experience”….

But this seems to get rid of spacetime for the events therein – as string theory attempts to do. Penrose OR has a structured spacetime and events therein

Reviewer 2 Report

The paper is very ambitious. It explicitly claims to address consciousness, but it never clarifies what notion is being used. Is it phenomenal consciousness but not access consciousness, transitive or intransitive? What about consciousness in relation to attention? Can attention be what the authors have in mind, and so could it happen without conscious awareness, as observers can attend without being phenomenally conscious according to many views of consciousness in psychology and neuroscience? The paper never talks about the Integrated Information Theory--a central theory of consciousness that explicitly addresses differences in information by employing an intrinsic instead of an extrinsic metric. These are all major omissions. 

The paper, however, is quite interesting. I cannot see clearly how the authors would revise it in order to engage the relevant literature and make more relevant to philosophers of mind, neuroscientists, or psychologists. Perhaps a discussion of how Shannon information applies to their framework? As it stands, it almost reads as a philosophy of physics paper, with no connection to standard views of consciousness in science. There are many recommendations I have for the bibliography, but what the paper needs now is clarity on what the scope of the paper is--anything on the distinctions between phenomenal consciousness, attention, or what makes attention conscious would be useful. This is because the paper can be read from a perspective in which phenomenal consciousness is not playing a key role. 

If the paper were on the limits of strictly externalist approaches to metaphysics and science, I would accept it without hesitation. But since it is about consciousness, and so much of the relevant literature is never mentioned or ignored, the paper must undergo major revisions for it to be acceptable.

Round 2

Reviewer 1 Report

I thank the authors for their responses, and do not wish to prolong the review process, as I agree this paper should be published. I do have a few more, relatively minor points however.

I also thank the authors for their recognition of the Penrose-Hameroff ‘Orch OR’ (orchestrated objective reduction) theory beginning bottom page 3, and recognize this paper is about their theory, not Orch OR.

However, it would be polite and informative to mention the name of the Orch OR theory, as, for example, IIT is mentioned by name several times (despite being

completely unrelated to neutral monism as far as I can tell). The name Orch OR explains the theory in a nutshell: OR (objective reduction) is Penrose’s view of self-collapse of the wavefunction which results in moments of proto-conscious experience, and Orch (orchestration) refers to the organization, integration and processing of quantum information, purportedly by brain microtubules prior to OR threshold, resulting in fully conscious moments, rather than merely random, disconnected proto-conscious ones. Orch also enables the results of the Orch OR event to be influenced by what Penrose calls non-computable Platonic values intrinsic to spacetime geometry, with possible deviation from Born rule randomness (a point that comes up in the paper).

The main point is that Orch OR is not ‘panpsychism’, and shouldn’t be lumped or dismissed accordingly. The phenomenal experience in Orch OR is in the quantum state reduction event which selects particular states of matter, rather than phenomenal experience being a property ofthe state of matter. Thus Orch OR is a process, consistent with Whitehead occasions. Each ‘collapse’ event is a moment of experience. Spacetime geometry is the underlying neutral entity which undergoes OR (or Orch OR) to produce 1) states of matter, and 2) moments of experience.

Regarding ‘Hilbert space realism’, Orch OR views superposition as separations in spacetime geometry. Hilbert space is a mathematical characterization. Its not Hilbert space that’s real, its spacetme geometry and its separations that are real.

And spacetime in this view is consistent with (as the authors state) nothing but the events therein (the events being OR events).

Please note that in the top paragraph on Page 4 Hameroff is mispelled as Hammerof.

The authors seem to claim their approach solves the problem of time, of the NOW. In Orch OR each Orch OR events is a moment of NOW, and a sequence of such gives rise to the flow of time.     

Regarding evolution, OR would have been present in the universe all along, and through pleasure, could have prompted life’s origin, and be driving its evolution.

See

The quantum origin of life - How the brain evolved to feel good. In On Human Nature, 2017 333-353 https://www.sciencedirect.com/science/article/pii/B978012420190300020X

I’m not asking the authors to ‘attack’ (nor to support) Orch OR, merely to recognize it. An added bonus would be if they could name a feature of their approach which is in any way better than those of Orch OR, a highly developed theory which specifies (in the framework of neutral monism) the underlying neutral monism entity (spacetime geometry), the reduction mechanism (OR by E-=h/t), the proto-conscious moments as the states of matter selected. Finally, does the authors’ approach entail quantum processes in the brain? Quantum biology? If so, how/where, and what about decoherence. Orch OR took a lot of abuse for many years before solving it.

Thanks for including references to Orch OR. However:

Penrose R, Hameroff S (1995) From supervenience to superdupervenience: Meeting the demands of a material world  J Consciousness Studies 2, 98-111

doesn't exist.

That clever title appears to be from a 1993 paper by Terence Horgan. Our paper JCS 2, 98-11 is our first paper, prompted by an attempted pre-emptive attack article by Pat Churchland and Rick Grush called (“Gaps in Penrose’s toilings”). Our response is called “ Gaps, what gaps?” You should check it out, at least for the cartoon at the end.

But the paper you should cite from back then actually lays out the neutral monism picture in Orch ORHameroff S, Penrose R (1996) Consciousness as orchestrated spacetime selections J Consciousness Studies3 (1)36-53.

You cite our 2014 ‘Reply to seven commentaries’ paper which is a supplement to our paper (which would be far better to cite)

Consciousness in the universe: Review of the Orch OR theory Physics of Life reviews 11(1):39-78 https://www.sciencedirect.com/science/article/pii/S1571064513001188

Orch OR is being tested, and is easily falsifiable. How would you test or falsify your Lagrangian approach?

Author Response

See attached response.

Reviewer 2 Report

I remain skeptical that the problem of consciousness can be scientifically solved with some equations or general statements about the universe. But this is not the authors' fault. One thing that would help the paper is if the authors presented a more balanced view, and did not take for granted the views of other authors as definitive authority. Some of the articles in this volume provide a more critical perspective, and would probably be helpful in balancing the article: de Barros and Montemayor, 2019, Quanta and Mind, Springer.

Author Response

See attached response.
